# Bacterial vampirism mediated through taxis to serum

**Siena J Glenn[1†], Zealon Gentry-Lear[2†], Michael Shavlik[2], Michael J Harms[2,3], Thomas J Asaki[4], Arden Baylink[1]***

[1]Washington State University, Department of Veterinary Microbiology and Pathology, Pullman, United States; [2]University of Oregon, Institute of Molecular Biology, Eugene, United States; [3]University of Oregon, Department of Chemistry & Biochemistry, Eugene, United States; [4]Washington State University, Department of Mathematics and Statistics, Pullman, United States

## eLife assessment

This work uses an interdisciplinary approach combining microfluidics, structural biology, and genetic analyses to provide **important** findings that show that pathogenic enteric bacteria exhibit taxis toward human serum. The data are **compelling** and show that the behavior utilizes the bacterial chemotaxis system and the chemoreceptor Tsr, which senses the amino acid L-serine. The work provides an ecological context for the role of serine as a bacterial chemoattractant and could have clinical implications for bacterial bloodstream invasion during episodes of gastrointestinal bleeding.

***For correspondence:**
arden.baylink@wsu.edu

†These authors contributed equally to this work

**Abstract** Bacteria of the family Enterobacteriaceae are associated with gastrointestinal (GI) bleeding and bacteremia and are a leading cause of death, from sepsis, for individuals with inflammatory bowel diseases. The bacterial behaviors and mechanisms underlying why these bacteria are prone to bloodstream entry remain poorly understood. Herein, we report that clinical isolates of non-typhoidal *Salmonella enterica* serovars, *Escherichia coli*, and *Citrobacter koseri* are rapidly attracted toward sources of human serum. To simulate GI bleeding, we utilized an injection-based microfluidics device and found that femtoliter volumes of human serum are sufficient to induce bacterial attraction to the serum source. This response is orchestrated through chemotaxis and the chemoattractant L-serine, an amino acid abundant in serum that is recognized through direct binding by the chemoreceptor Tsr. We report the first crystal structures of *Salmonella* Typhimurium Tsr in complex with L-serine and identify a conserved amino acid recognition motif for L-serine shared among Tsr orthologues. We find Tsr to be widely conserved among Enterobacteriaceae and numerous World Health Organization priority pathogens associated with bloodstream infections. Lastly, we find that Enterobacteriaceae use human serum as a source of nutrients for growth and that chemotaxis and the chemoreceptor Tsr provide a competitive advantage for migration into enterohemorrhagic lesions. We define this bacterial behavior of taxis toward serum, colonization of hemorrhagic lesions, and the consumption of serum nutrients as 'bacterial vampirism', which may relate to the proclivity of Enterobacteriaceae for bloodstream infections.

## Introduction

Bacteria use chemosensory systems to survey and navigate their environments (*Gumerov et al., 2021*; *Wadhwa and Berg, 2022*; *Zhou et al., 2023*). In the exceptionally dynamic environment of the host gut, peristalsis and flow constantly perturb the microscopic physicochemical landscape. Responding to such transient and shifting stimuli is enabled by chemosensing that allows bacteria to rapidly

**eLife digest** Sepsis is the leading cause of death in patients with inflammatory bowel disease. Individuals with this condition can experience recurrent episodes of intestinal bleeding, giving intestinal (or enteric) bacteria an entry point into the bloodstream. This puts patients at risk of developing fatal infections – particularly from infections caused by bacteria belonging to the Enterobacteriaceae family. However, it is not well understood why this family of bacteria are particularly prone to entering the bloodstream.

Enteric bacteria commonly respond to chemicals (or chemical stimuli) in their environment. This process, known as chemotaxis, helps bacteria with a variety of tasks, such as monitoring their environment, moving to different areas within their environment or colonizing their host. Chemical stimuli are classed as 'attractants' or 'repellents', with attractants luring the bacteria to an area and repellents discouraging the bacteria from being in a specific place.

Intestinal bleeds will release serum (the liquid part of blood) into the gut, which could serve as a source of chemical stimuli to attract Enterobacteriaceae into the bloodstream. To find out more, Glenn, Gentry-Lear et al. first used a microfluidic device to simulate an intestinal bleed and tested the response of Enterobacteriaceae bacteria to serum. Using chemotaxis, bacteria were found to be attracted to the amino acid L-serine in the serum to which they were able to attach through a receptor called Tsr. They also consumed nutrients present in the human serum to help them grow. Experiments with intestinal tissue showed that chemotaxis attracted bacteria to bleeding blood vessels and the Tsr receptor helped them to infiltrate the blood vessels. Glenn et al. termed this attraction to and feeding upon blood serum as 'bacterial vampirism'.

These findings suggest that chemotaxis of Enterobacteriaceae towards L-serine in serum may be linked to their tendency to enter the bloodstream. Developing therapies that target chemotaxis in Enterobacteriaceae may provide a method for managing bloodstream infections.

restructure their populations, within seconds, through taxis toward or away from effector sources (*Huang et al., 2017*; *Huang et al., 2015*; *Perkins et al., 2019*). Many enteric pathogens and pathobionts use a form of chemosensing known as chemotaxis to colonize specific tissues, whereby bacteria swim toward or away from sources of exogenous nutrients, toxins, and host-emitted cues (*Zhou et al., 2023*; *Keegstra et al., 2022*; *Matilla and Krell, 2018*). Previously, we determined that most bacterial genera classified by the World Health Organization (WHO) as 'priority pathogens' employ chemotaxis systems for efficient infection and exert precise control over their colonization topography (*Zhou et al., 2023*; *Murray et al., 2022*; *Tacconelli et al., 2018*). These include multidrug-resistant Enterobacteriaceae pathogens of the gut such as *Salmonella enterica*, Enterohemorrhagic *Escherichia coli* (EHEC), *Citrobacter koseri*, and *Enterobacter cloacae*, which present major challenges for nosocomial infections and global health (*Zhou et al., 2023*; *Tacconelli et al., 2018*).

Mapping the chemosensing signals enteric bacteria perceive within the host can provide insight into pathogenesis and perhaps be a source of drug targets for inhibiting pathogen navigation and colonization (*Zhou et al., 2023*; *Matilla and Krell, 2023*). Gut dysbiosis—a diseased host state induced by pathologies, inflammation, and infections—exposes enteric bacteria to chemical stimuli distinct from that of a healthy gut that could encourage opportunistic pathogenesis (*Zhou et al., 2023*; *Belizário and Faintuch, 2018*). Yet, the stimuli sensed by bacterial pathogens and pathobionts within the dysbiotic gut environment remain poorly understood. Investigating how chemosensing controls the responses of pathogen populations to chemical features associated with dysbiosis presents an opportunity to gain deeper insights into the critical juncture between infection resolution and exacerbation (*Figure 1*; *Zhou et al., 2023*; *Belizário and Faintuch, 2018*; *Rogers et al., 2020*; *Donaldson et al., 2016*).

In a diseased gut, enteric bacteria may encounter a distinctive host-derived chemical feature not found in a healthy gut: GI bleeding. The microenvironment of an enteric hemorrhagic lesion involves a source of serum, the liquid component of blood, emanating from the host tissue and diffusing into the intestinal lumen (*Figure 1*). In effect, this creates a microscopic gradient, that is, a microgradient, of chemicals flowing outward from a point source that may serve as chemosensory signals for bacteria. Enteric infections caused by Enterobacteriaceae species can lead to GI bleeding, a condition with

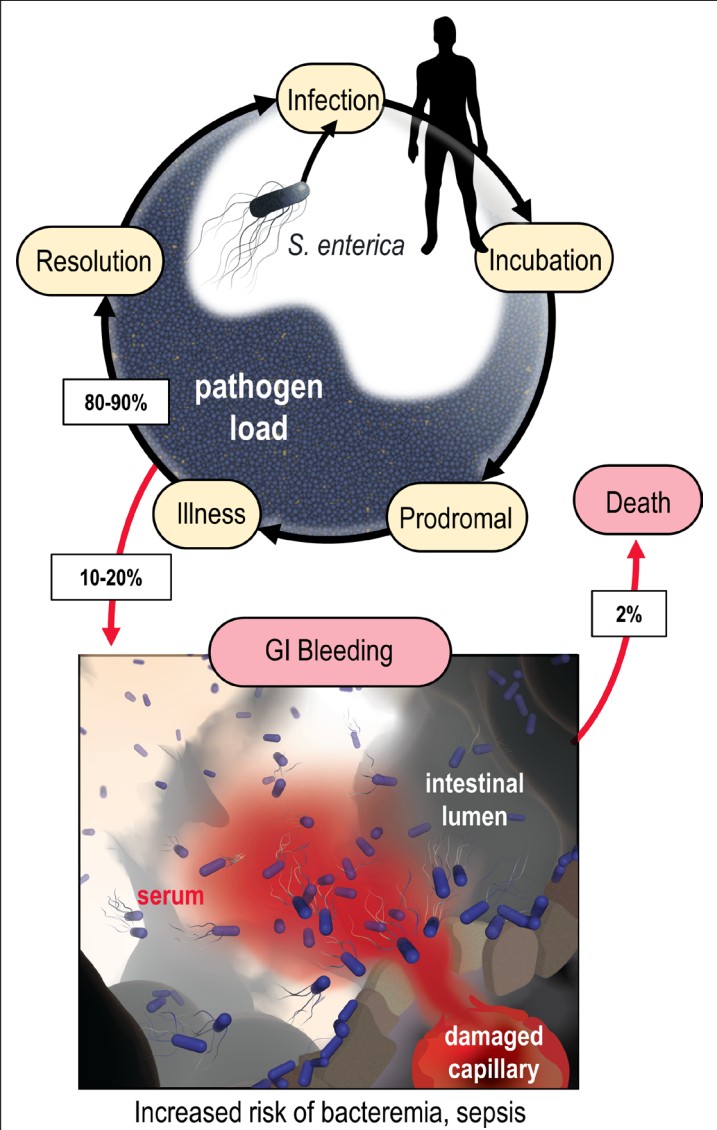

**Figure 1.** Model of the microenvironment of bacterial-induced hemorrhagic lesions. The typical course of non-typhoidal *S. enterica* infections is shown proceeding through infection, incubation, prodromal, illness, and resolution stages (black arrows). The atypical route of gastrointestinal (GI) bleeding, associated with increased mortality and morbidity, is shown in red arrows, with rates approximated from available literature. An artistic depiction of bacterial injury tropism is shown bottom.

a significant risk of mortality. Although severe GI bleeding is not a typical outcome of enteric infection, it is a substantial burden on human health, affecting around 40–150 out of every 100,000 individuals annually, with a fatality rate ranging from 6% to 30% (*Figure 1*; *Zhou et al., 2023*; *Tacconelli et al., 2018*; *Aljarad and Mobayed, 2021*; *Huang et al., 2021*; *Kim et al., 2014*; *Kim et al., 2014*; *Moledina and Komba, 2017*; *Saltzman, 2022*; *Scales and Huffnagle, 2013*). Enterobacteriaceae are prone to bloodstream entry and are a leading cause of sepsis-related deaths in individuals with inflammatory bowel diseases (IBDs) that have recurrent enterohemorrhagic lesions (*Weber et al., 2020*; *Goren et al., 2020*). Intestinal intra-abdominal abscesses, microperforations, and fistulas associated with IBD predispose patients to GI bleeding and bacteremia (*Satoh et al., 2020*; *Vohra et al., 2020*). Despite the established connection between Enterobacteriaceae-induced sepsis and GI bleeding, it remained unknown whether these bacteria perceive serum through chemosensing.

Serum is a complex biological solution with components that may enhance, or hinder, bacterial growth. It offers a rich reservoir of nutrients for bacteria: sugars and amino acids at millimolar concentrations and essential metals like iron and zinc (*Zhou et al., 2023*; *Murdoch and Skaar, 2022*). Yet, serum also contains host factors that inhibit bacterial proliferation in the bloodstream such as cathelicidin, defensins, and the complement system (*Cheng et al., 2019*; *Xu and Lu, 2020*). Consequently, how enteric pathogens and pathobionts might respond to serum diffusing into a liquid environment remained unclear. To address this open question, we elucidated how Enterobacteriaceae species use chemosensing to respond to serum, the molecular mechanism of this response, and how chemosensing is employed to enter and migrate into enterohemorrhagic lesions. Across all examined scenarios we observe these bacteria to exhibit remarkable sensitivity and attraction toward human serum. These findings suggest that environmental stimuli unique to the dysbiotic gut, sensed through bacterial chemosensory systems, can encourage pathogenic behaviors and adverse consequences for the host.

## Results

### Use of the chemosensory injection rig assay (CIRA) to study polymicrobial chemosensing behaviors

To model features of enteric bleeding in vitro, we utilized an experimental system to inject minute quantities of human serum into a pond of motile bacteria and observe real-time responses by microscopy (*Figure 2A*). The system and methodology, which we refer to herein as the CIRA, offer several

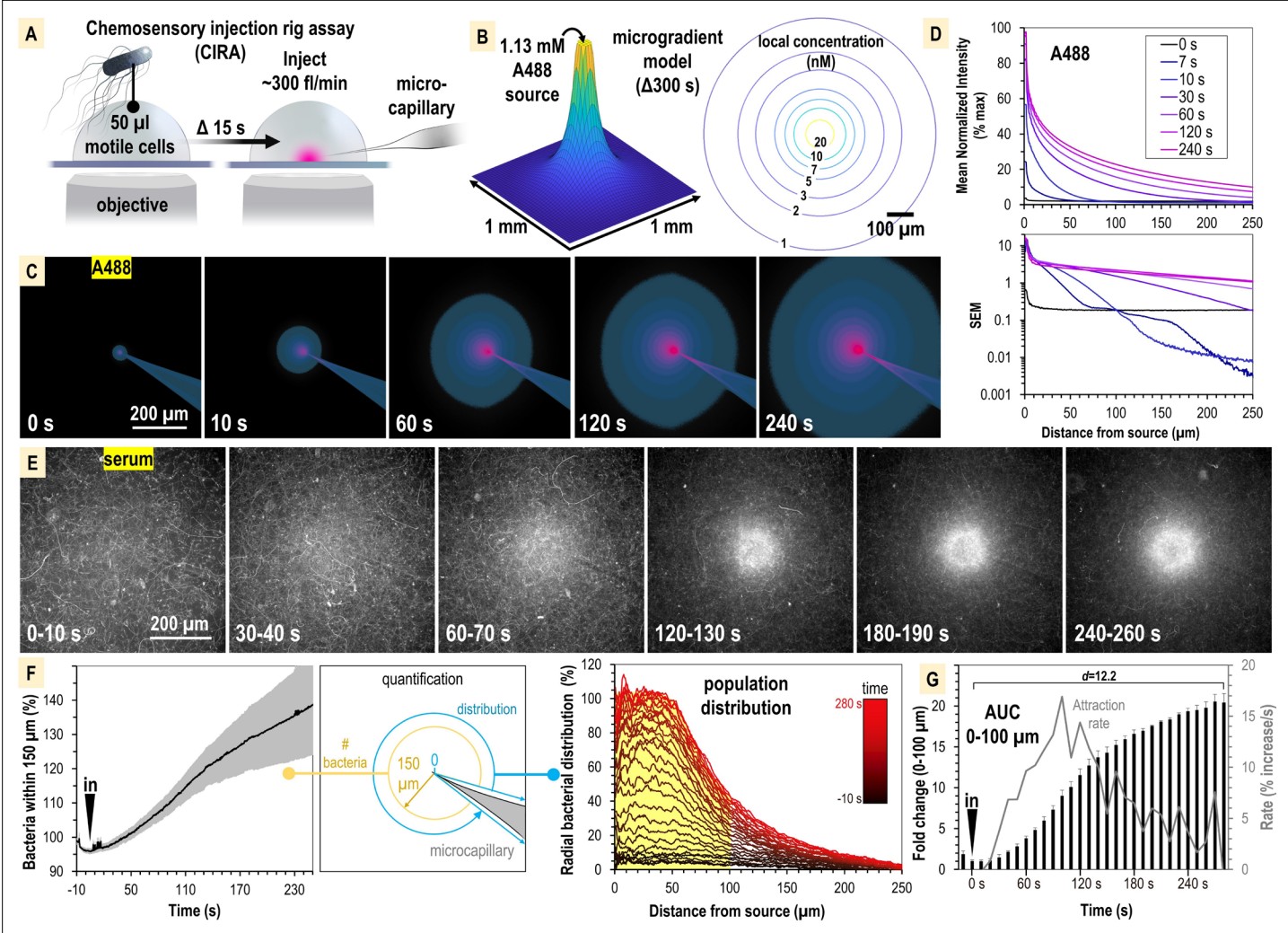

**Figure 2.** *S. enterica Typhimurium* IR715 rapidly localizes toward human serum. (**A**) Chemosensory injection rig assay (CIRA) experimental design. (**B**) CIRA microgradient diffusion model, simulated with a source of 1.13 mM A488 dye after 300 s of injection. (**C**) Experimental visualization of the CIRA microgradient with A488 dye. (**D**) Injection and diffusion of A488 dye. Shown at the top is the mean normalized fluorescence intensity at representative time points as a function of distance from the source, and shown at the bottom is the standard error of the mean (SEM) for these data (n = 6). (**E**) Response of *S. enterica* Typhimurium IR715 to human serum (max projections over 10 s intervals). (**F**) Quantification of *S. enterica* Typhimurium IR715 attraction response to human serum (n = 4, 37°C) characterized as either the relative number of bacteria within 150 µm of the source (left) or the radial distribution of the bacterial population over time (right, shown in 10 s intervals). (**G**) Area under the curve (AUC) versus time for the bacterial population within 100 µm of the serum treatment source (area indicated in yellow in **F**). Effect size (Cohen's *d*) between the treatment start and endpoints is indicated. Insertion of the treatment microcapillary is indicated with black 'in' arrow. Attraction rate over time indicated in gray. Data shown are means, error bars indicate SEM.

The online version of this article includes the following figure supplement(s) for figure 2:

**Figure supplement 1.** Chemosensory injection rig assay (CIRA) experimentation controls and microgradient modeling.

**Figure supplement 2.** Comparison of chemosensory injection rig assay (CIRA) diffusion modeling with experimental A488 data.

advantages for studying bacterial chemosensing and localization in response to serum: (1) we can use bona fide human serum, (2) the readouts are direct measurements of real-time localization dynamics of the bacterial population, and (3) similar to a bleeding event, a source of fresh serum is continuously emitted. By employing multichannel fluorescence imaging of differentially labeled bacterial populations, we observe polymicrobial interactions through head-to-head comparisons of bacterial behavior within the same experiment.

The CIRA microgradient is established through injection at a constant rate of ~300 fl/min, through a 0.5 µm glass microcapillary, and expands over time through diffusion (***Figure 2A and B***,

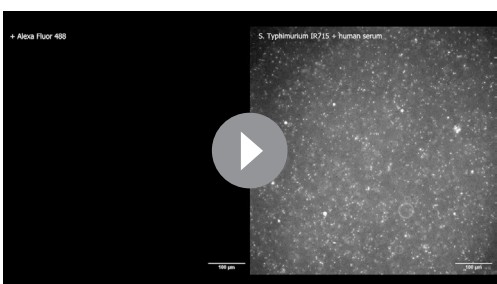

**Video 1.** A488 vs. *S. enterica* Typhimurium IR715 + serum. Representative chemosensory injection rig assay (CIRA) experiments with Alexa Fluor 488 dye (left) and *S. enterica* Typhimurium IR715 treated with human serum (right). Video is shown at 1× speed and is also viewable at https://www.youtube.com/watch?v=dyrQT2Ni5J8.

https://elifesciences.org/articles/93178/figures#video1

*Figure 2—figure supplement 1*). The diffusion of serum metabolites can be reasonably approximated by the three-dimensional differential diffusion equation (*Figure 2B*, *Figure 2—figure supplements 1 and 2* , 'Method details'). We calculate that the introduction of a minute volume of novel effector produces a steep microgradient, in which a millimolar source recedes to nanomolar concentrations after diffusion across only a few hundred microns (*Figure 2B*, *Figure 2—figure supplement 1*). For instance, in our assay we expect a bacterium 100 μm from a 1 mM infinite-volume source to experience a local concentration of a small molecule effector (diffusion coefficient approximately $4 \times 10^{-6}$ cm$^2$ s$^{-1}$) of 10 nM after 300 s of injection (*Figure 2B*, *Figure 2—figure supplement 1*). To visualize the microgradient experimentally, we utilized Alexa Fluor 488 dye (A488) and observed the microgradient to be stable and consistent across replicates (*Figure 2C and D*, *Video 1*). As a test case of the accuracy of the microgradient modeling, we compared our calculated values for A488 diffusion to the normalized fluorescence intensity at time 120 s. We determined the concentration to be accurate within 5% over the distance range 70–270 μm (*Figure 2—figure supplement 2*). At smaller distances (<70 μm), the measured concentration is approximately 10% lower than that predicted by the computation. This could be due to advection effects near the injection site that would tend to enhance the effective local diffusion rate. Additionally, we found no behavioral differences in treatments with buffer in the range of pH 4–9, indicating small, localized pH changes are inconsequential for taxis in our system, and that any artifactual forces, such as flow, account for only minor changes to bacterial distribution at the population level, in the range of ±10% (*Figure 2—figure supplement 1*).

## Non-typhoidal *S. enterica* serovars exhibit rapid attraction to human serum

We first studied the chemosensing behaviors of *S*. Typhimurium IR715, which is a derivative of ATCC 14028 originally isolated from domesticated chickens, and is used extensively in the study of *Salmonella* pathogenesis (see Key resources table; *Tsolis et al., 1999*; *Rivera-Chávez et al., 2016*; *Winter et al., 2010*; *Rivera-Chávez et al., 2013*). As a treatment, we utilized commercially available human serum that was not heat-inactivated nor exposed to chemical or physical treatments that would be expected to alter its native complement properties (see 'Materials and methods'). We assessed the response of motile IR715, containing a fluorescent mPlum marker, to a source of human serum over the course of 5 min (*Figure 2E*, *Video 1*). During this timeframe, we witnessed a rapid attraction response whereby the motile bacterial population reorganized from a random distribution to one concentrated within a 100–150 μm radius of the serum source (*Figure 2E and F*, *Video 1*). To compare responses between CIRA experiments, we plotted data as either the relative number of bacteria within 150 μm of the treatment (*Figure 2F*, left panel) or a radial distribution of the population (*Figure 2F*, right panel) over time. By these metrics, we determined *S*. Typhimurium IR715 is attracted to human serum. The bacterial population in proximity to the serum source doubles by 40 s, reaches a maximal attraction rate by 90 s, and approaches equilibrium by 300 s post-treatment (*Figure 2G*).

To test whether serum attraction is observed in *Salmonella* strains that infect humans, and if responses differ among non-typhoidal *Salmonella* serovars, we employed dual-channel CIRA imaging to compete *S*. Typhimurium IR715 against representative clinical isolates of Typhimurium (SARA1), Newport (M11018046001A), and Enteriditis (05E01375). These serovars are the most common in North American infections (*Ferrari et al., 2019*). In varying magnitude, all strains showed attraction responses to human serum, seen as a significant distribution bias toward the serum source relative to the experiment periphery (*Figure 3*, *Video 2*). We noted that in some experiments the population

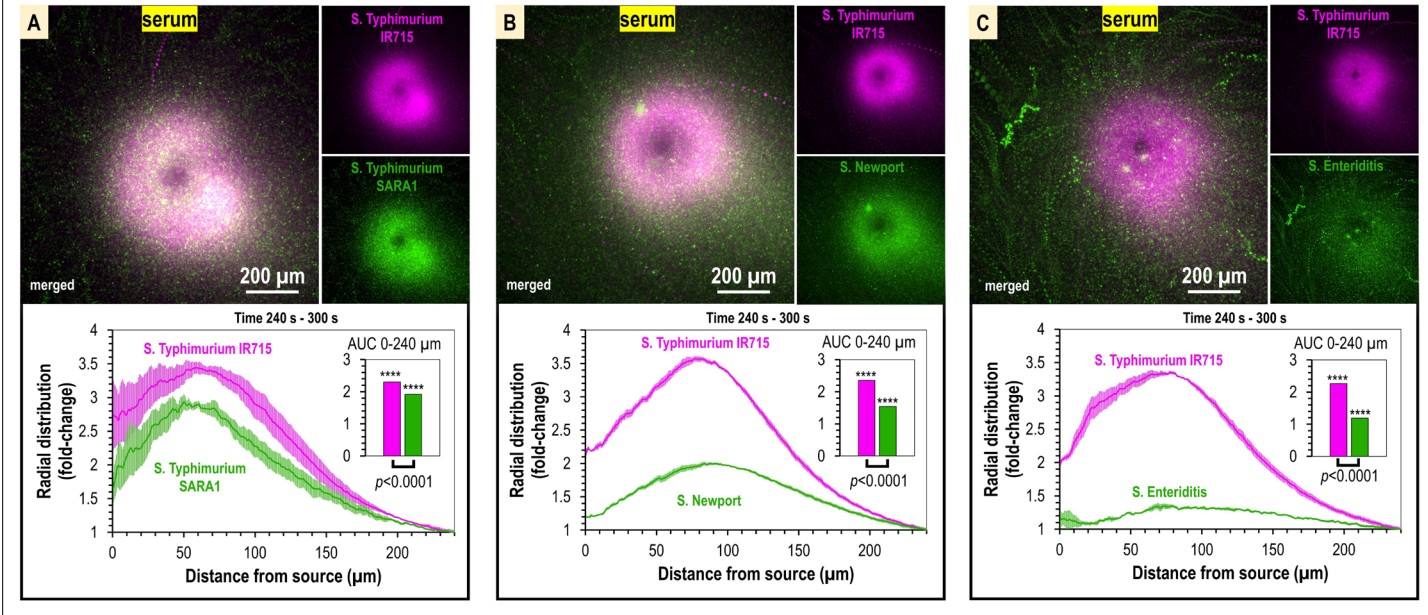

**Figure 3.** Taxis to human serum is retained across *S. enterica* clinical isolates and diverse serovars. (**A–C**) Chemosensory injection rig assay (CIRA) competition experiments between *S.* Typhimurium IR715 (pink) and clinical isolates (green) responding to human serum for 5 min (n = 4, 37°C). Images are representative max projections over the final minute of treatment. Radial distributions calculated from max projections and averaged across replicates are shown as fold-change relative to the image periphery at 240 μm from the source. Inset plots show fold-change area under the curve (AUC) of strains in the same experiment relative to an expected baseline of 1 (no change). p-Values shown are calculated with an unpaired two-sided *t*-test comparing the means of the two strains, or one-sided *t*-test to assess statistical significance in terms of change from onefold (stars). Data shown are means, error bars indicate SEM.

peak is 50–75 μm from the source, possibly due to a compromise between achieving proximity to nutrients in the serum and avoidance of bactericidal serum elements, but this behavior was not consistent across all experiments. Overall, our data show *S. enterica* serovars that cause disease in humans are exquisitely sensitive to human serum, responding to femtoliter quantities as an attractant, and that distinct reorganization at the population level occurs within minutes of exposure (*Figure 3*, *Video 2*).

## Chemotaxis and the chemoreceptor Tsr mediate serum attraction

Serum is a complex biological solution that contains sugars, amino acids, and other metabolites that could serve as attractant signals (*Zhou et al., 2023*). Based on the rapid attraction of motile, swimming bacteria to the treatment source, characteristic of chemotactic behaviors (*Huang et al., 2015*), we hypothesized that one or more of these chemical components are specifically recognized as chemoattractants through the repertoire of chemoreceptors possessed by *Salmonellae* (*Figure 4A*). Based on known chemoreceptor–chemoattractant ligand interactions (*Zhou et al., 2023*; *Matilla et al., 2022*), we identified three chemoreceptors that might mediate taxis toward serum: (1) taxis to serine and repellents (Tsr), which responds to L-serine, and reportedly also norepinephrine (NE) and 3,4-dihydroxymandelic acid (DHMA); (2) taxis to ribose and glucose/galactose (Trg), which responds to glucose and galactose; and (3) taxis to aspartate and repellents (Tar), which responds to L-aspartate (*Figure 4A*). We modeled the local concentration profile of these effectors based on their typical concentrations in human serum (*Figure 4B*). Of these, by far the two most prevalent chemoattractants in serum are glucose

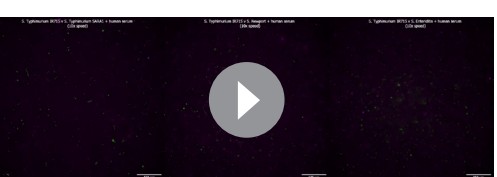

**Video 2.** Comparison of responses to human serum between *Salmonella* strains. Representative chemosensory injection rig assay (CIRA) experiments comparing responses to human serum between *S. enterica* Typhimurium IR715 (*mplum*) and clinical isolates (*gfp*), as indicated. Videos depict responses over 5 min of treatment. Viewable at https://youtu.be/dwtZtoisjrU.

https://elifesciences.org/articles/93178/figures#video2

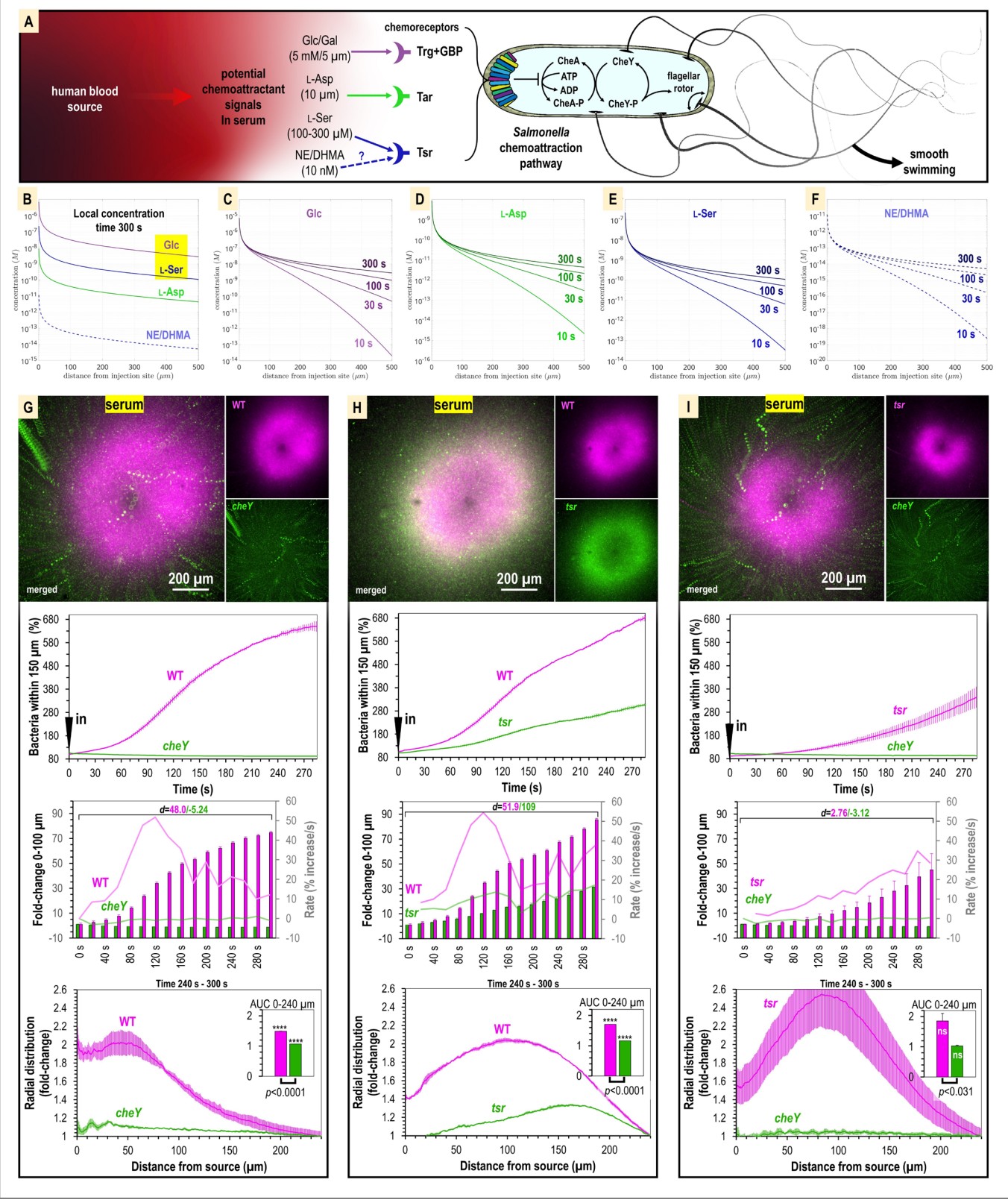

**Figure 4.** Attraction to human serum is mediated through chemotaxis and the chemoreceptor Tsr. (**A**) Potential mechanisms involved in *Salmonella* sensing of chemoattractants present in human serum. Approximate concentrations of these effectors in human serum are indicated in parentheses (*Zhou et al., 2023*). (**B–F**) Microgradient modeling of serum chemoattractant concentrations. (**G–I**) Chemosensory injection rig assay (CIRA) competition experiments between *S.* Typhimurium IR715 WT and isogenic mutants *cheY*, or *tsr*, and *tsr* versus *cheY*, in response to human serum (n = 3–4, 37°C).

*Figure 4 continued on next page*

*Figure 4 continued*

Rates in terms of fold-change are indicated with light pink/light green lines and plotted on the gray secondary y-axis. Inset area under the curve (AUC) plots are shown as described in *Figure 3*. Data are means, and error bars are SEM.

The online version of this article includes the following figure supplement(s) for figure 4:

**Figure supplement 1.** Comparison of *S.* Typhimurium IR715 chemotactic responses to human serum, human serum with serine racemase treatment, L-serine versus 3,4-dihydroxymandelic acid (DHMA) and norepinephrine (NE) .

(5 mM) and L-serine (100–300 µM) (*Figure 4B–F*). This suggested to us that the chemoreceptors Trg and/or Tsr could play important roles in serum attraction. These chemoreceptors were also previously shown to provide colonization advantages during *S.* Typhimurium infection (*Rivera-Chávez et al., 2013*).

To test the role of chemotaxis in serum attraction, we competed wildtype (WT) *S.* Typhimurium IR715 against a chemotaxis-null isogenic *cheY* mutant, which possesses swimming motility but is blind to chemoeffector signals (*Figure 4G*, *Video 3*). Whereas the WT mounts a robust attraction response to serum, the *cheY* mutant population remains randomly distributed (*Figure 4G*, *Video 3*). We also observed the *cheY* mutant to exhibit a slight decline in cells proximal to the treatment source over time, which we attribute to cellular crowding effects from the influx of WT cells (*Figure 4G*, *Video 3*). In this background, the fraction of WT cells within 100 µm of the serum source increases by 70-fold, with the maximal rate of attraction achieved by 120 s post-treatment. Thus, we determined that chemotaxis is responsible for the rapid localization of *S. enterica* to human serum.

We next analyzed strains with deletions of the chemoreceptor genes *trg*, or *tsr*, to test the roles of these chemoreceptors in mediating taxis to serum. We were surprised to find that the *trg* strain had deficiencies in swimming motility (data not shown). This was not noted in earlier work but could explain the severe infection disadvantage of this mutant (*Rivera-Chávez et al., 2013*). Because motility is a prerequisite for chemotaxis, we chose not to study the *trg* mutant further and instead focused our investigations on Tsr. We compared chemoattraction responses in dual-channel CIRA experiments between the WT and *tsr* mutant and observed an interesting behavior whereby both strains exhibited chemoattraction, but the *tsr* mutant distribution was relegated to a halo at the periphery of the WT peak (*Figure 4H*, *Video 3*). The WT efficiently outcompetes the *tsr* mutant such that by 5 min post-treatment the ratio of WT to *tsr* cells proximal to the serum source is 3:1 (*Figure 4H*, *Video 3*). Similar to *cheY*, we presume the *tsr* halo results from cellular crowding effects induced by the high density of WT cells near the serum source. To test how the *tsr* mutant responds to serum in the absence of a strong competitor, we compared chemoattraction between *tsr* and *cheY*. In this background, *tsr* chemoattraction remained diminished relative to that of WT, but the *tsr* distribution shifted closer to the serum source (*Figure 4I*, *Video 3*). Since *tsr* mutation diminishes serum attraction but does not eliminate it, we conclude that multiple chemoattractant signals and chemoreceptors mediate taxis to serum. To further understand the mechanism of this behavior, we chose to focus on Tsr as a representative chemoreceptor involved in the response, presuming that serum taxis involves one, or more,



**Video 3.** Comparison of responses to human serum between wildtype (WT) and chemotactic mutants (IR715 background). Representative chemosensory injection rig assay (CIRA) experiments comparing responses to human serum between WT *S. enterica* Typhimurium IR715 (*mplum* in left and center panel) and chemotactic mutants *tsr* (*gfp* in center panel, *mplum* in right panel), and *cheY* (*gfp* in left and right panel). Videos depict responses over 5 min of treatment. Viewable at https://youtu.be/O5zsEAqcJw8.

https://elifesciences.org/articles/93178/figures#video3

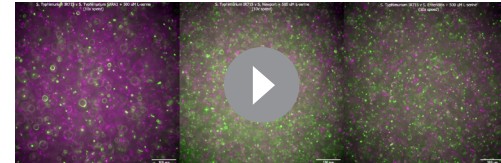

**Video 4.** Comparison of responses to 500 µM L-serine between *Salmonella* strains. Representative chemosensory injection rig assay (CIRA) experiments comparing responses to 500 µM L-serine between *S. enterica* Typhimurium IR715 (*mplum*) and clinical isolates (*gfp*), as indicated. Videos depict responses over 5 min of treatment. Viewable at https://www.youtube.com/watch?v=p0Tsp06ZHO8.

https://elifesciences.org/articles/93178/figures#video4

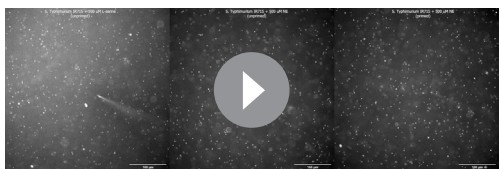

**Video 5.** Comparison of responses of *S.* Typhimurium IR715 to L-serine and norepinephrine. Chemosensory injection rig assay (CIRA) experiments comparing response of *S.* Typhimurium IR715 to L-serine or to norepinephrine. Cells are unprimed or primed with norepinephrine (NE), as indicated. Videos depict responses over 5 min of treatment. Viewable at https://youtu.be/pUOlVjKYptc.

https://elifesciences.org/articles/93178/figures#video5

of the chemoattractants recognized by Tsr that is present in serum: L-serine, NE, or DHMA.

## *S.* Typhimurium exhibits chemoattraction to L-serine, but not NE or DHMA

We next sought to identify the specific chemoattractants driving Tsr-mediated serum attraction. Suspecting that one effector might be L-serine, we took advantage of the fact that Tsr can only bind the L and not the D enantiomer and treated the serum with serine racemase, an enzyme that converts L- to D-serine. The attraction of *S.* Typhimurium IR715 to serum treated with serine racemase is diminished and the population is more diffuse (*Figure 4—figure supplement 1A and B*).

In competition experiments between WT and chemotaxis-deficient strains, the WT cells proximal to the treatment source are reduced by about half (*Figure 4—figure supplement 1C*), and WT no longer possesses an attraction advantage over the *tsr* mutant (*Figure 4—figure supplement 1D*). These data support that the deficiency of the *tsr* strain in serum taxis is due to an inability to sense L-serine within serum.

We next used CIRA to examine responses to purified effectors diluted in buffer. The concentration of L-serine in human serum ranges from approximately 100–400 μm, depending on diet and health factors (*He et al., 2022*; *Miller et al., 2020*; *Pitkänen et al., 2003*), whereas the neurotransmitter NE, and its metabolized form, DHMA, are thought to circulate at approximately 10 nM (*Kopin et al., 1978*; *Steuer et al., 2016*). It has been proposed that, like L-serine, NE and/or DHMA are sensed directly by Tsr and at even higher (nanomolar) affinity (*Orr et al., 2020*; *Pasupuleti et al., 2014*). We used CIRA to test the response of *S.* Typhimurium IR715 to L-serine, NE, and DHMA. We observed robust chemoattraction responses to L-serine, evident by the accumulation of cells toward the treatment source (*Figure 4—figure supplement 1E*, *Video 4*), but no response to NE or DHMA, with the cells remaining randomly distributed even after 5 min of exposure (*Figure 4—figure supplement 1F–I*, *Videos 5 and 6*). Following extensive investigation and adjusting experiment parameters such as culture protocol, cell density, temperature, injection flow, and compound concentration, we saw no evidence that NE or DHMA are chemoeffectors for *S.* Typhimurium (*Figure 4—figure supplement 1F–I*, *Videos 5 and 6*). Since this result is surprising given previous reports (*Orr et al., 2020*; *Pasupuleti et al., 2014*), we have provided a database of 17 videos of CIRA experiments showing the null response to DHMA and NE, versus L-serine, under various conditions (https://public.vetmed.wsu.edu/Baylink). Together, our data indicate the chemoattractant component of serum that is sensed through Tsr is L-serine.

Chemoattraction to L-serine has mostly been studied in the context of model laboratory strains

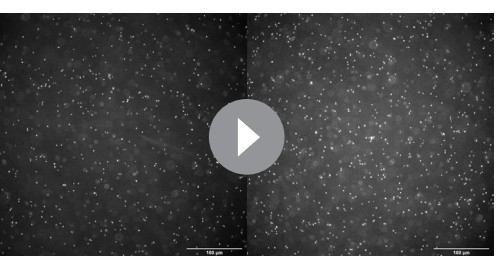

**Video 6.** Chemosensory injection rig assay (CIRA) experiments showing response of *S.* Typhimurium IR715 to DHMA. Cells are unprimed (left) or primed with norepinephrine (NE) (right). Viewable at https://youtu.be/j4YL95QFCuI.

https://elifesciences.org/articles/93178/figures#video6

and has not been rigorously evaluated for *S. enterica* clinical isolates or various serovars. To establish whether L-serine sensing is observed in strains responsible for human infections, we used dual-channel CIRA to compare chemoattraction to L-serine between *S.* Typhimurium IR715 and clinical isolates (*Figure 5A–C*, *Video 4*). In each case, we observe robust chemoattraction, though there are differences in sensitivity to L-serine. The magnitude of chemoattraction was highest for *S.* Typhimurium SARA1 and *S.* Newport, whereas *S.* Typhimurium IR715 and *S.* Enteriditis showed lower responses, which could relate to the different host specificities of these serovars and strains (*Figure 5A–C*, *Video 4*).

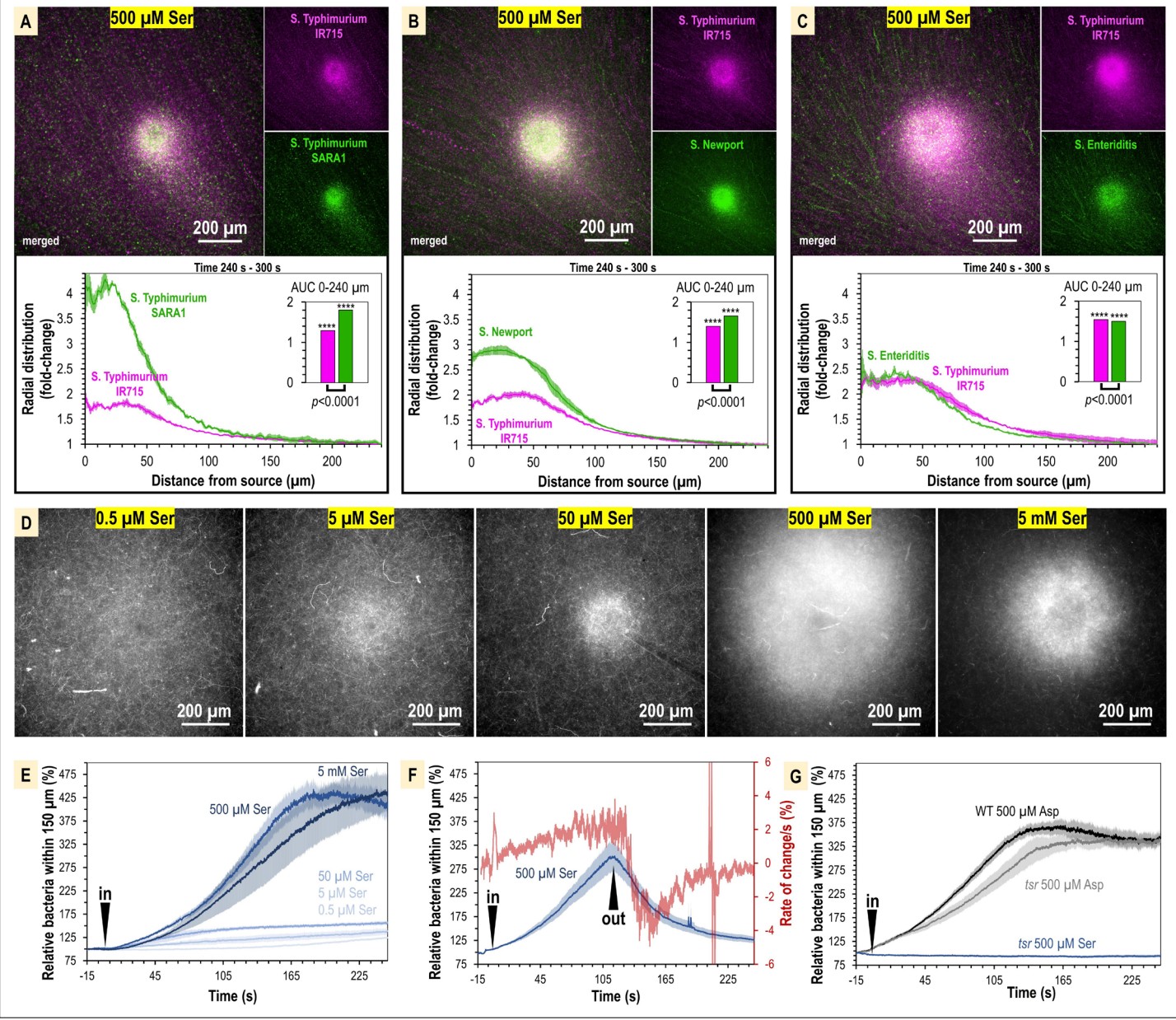

**Figure 5.** The concentration of L-serine in human serum is sufficient to mediate chemoattraction. (**A–C**) Chemosensory injection rig assay (CIRA) competition experiments between *S.* Typhimurium IR715 (pink) and clinical isolates (green) in response to 500 µM L-serine (n = 3, 37°C). (**D**) Representative results showing max projections of *S.* Typhimurium IR715 at 240–300 s post CIRA treatment with L-serine concentrations (30°C). (**E**) Quantification of multiple replicate experiments shown in (**D**). (**F**) Attraction and dispersion of *S.* Typhimurium IR715 following addition and removal of 500 µM L-serine source (30°C). (**G**) *S.* Typhimurium IR715 WT or *tsr* mutant responses to L-aspartate or L-serine treatments (30°C).

The online version of this article includes the following figure supplement(s) for figure 5:

**Figure supplement 1.** Calculation of L-serine source required for half-maximal chemoattraction response ($K_{1/2}$).

We next used CIRA to test a range of L-serine sources spanning five orders of magnitude to define whether the concentrations of L-serine present in human serum are sufficient to drive a chemoattraction response. Within the 5 min timeframe of our experiments, the minimal source concentration of L-serine needed to induce chemoattraction is 0.5–5 µM. The [L-serine] source required for half-maximal chemoattraction (i.e., $K_{1/2}$) is approximately 105 µM (*Figure 5D and E*, *Figure 5—figure supplement 1*). Based on our microgradient modeling, this corresponds to a local L-serine concentration of 1 nM for bacteria 100 µm from the source at t = 300 s (*Figure 2—figure supplement 1*,

*Figure 4B*). Therefore, we determined that the typical concentration of L-serine in human serum is >200-fold greater than the minimum required to elicit chemoattraction. To gain further insights into the dynamics of L-serine chemoattraction, we monitored chemotactic behavior in the presence of L-serine, and then removed the treatment (*Figure 5F*). These experiments showed maximal attraction and dispersal rates to be similar, changing by approximately 4% per second (*Figure 5F*). These findings emphasize the rapid dynamics through which chemotaxis can influence the localization of bacteria in response to microscopic gradients of chemoeffectors.

To substantiate our findings, we considered some alternative explanations for our data. First, we tested whether Tsr alone was required for chemoattraction to L-serine, or whether some other chemoreceptor, or form of taxis, such as energy or redox taxis, might contribute to the responses. However, we found when treated with L-serine, the *tsr* mutant showed no chemoattraction and behaved similarly to the chemotaxis-null *cheY* mutant (*Figure 5G*). Second, we considered the possibility that the defect in serum attraction of the *tsr* mutant could be due to pleiotropic effects of the *tsr* deletion on chemotaxis signaling or motility, similar to the inhibited swimming motility of the *trg* mutant. We tested the ability of the *tsr* strain to respond to another chemoattractant, L-aspartate, which is sensed through the Tar chemoreceptor (*Figure 4A*). We found that the *tsr* mutant mounted a robust chemoattraction response to 500 µM L-aspartate, similar in magnitude and rate to WT (*Figure 5G*), supporting that chemotaxis to non-serine stimuli remains functional in the *tsr* strain. Along with the data showing the WT attraction to serum is diminished with serine racemase treatment, these results support that the mechanism of Tsr-mediated chemoattraction to serum is through direct recognition of L-serine.

## Serum provides a growth advantage for non-typhoidal *S. enterica* serovars

The robust serum attraction response conserved across diverse *Salmonella* serovars suggests serum, and L-serine present in serum, could be a source of nutrients during infection. Yet, serum also contains bactericidal factors that could inhibit bacterial growth (*Cheng et al., 2019*). To address the uncertainty as to whether serum would benefit or inhibit bacterial growth, we added human serum to *Salmonella* in liquid culture grown on minimal media. In all strains surveyed, we found that serum addition enhances growth, and we saw no evidence of killing even for the highest serum concentrations tested (*Figure 6*). The growth enhancement requires relatively little serum, as 2.5% v/v is sufficient to provide a 1.5–2.5-fold growth increase (*Figure 6*, https://public.vetmed.wsu.edu/Baylink).

L-serine is not only a chemoattractant, but also an important nutrient for bacteria in the gut (*Zhou et al., 2023*), so we hypothesized the growth benefit could be from L-serine. We determined by mass spectrometry that our human serum samples contain 241 µM ± 48 total serine (L- and D-enantiomers), of which approximately 99% is expected to be L-serine (*Hashimoto et al., 2003*; *Figure 6—figure supplement 1*). We attempted to treat human serum with a purified recombinant enzyme that degrades L-serine, serine dehydrogenase (SDS), to see whether serine-depleted serum would elicit less growth or chemoattraction. However, SDS treatments did not alter serum serine content so we abandoned this approach (*Figure 6—figure supplement 1*). Instead, we performed a titration of purified L-serine and assessed its role in supporting a growth advantage. We found that only a very small benefit is achieved with the addition of L-serine, which did not recapitulate the larger growth benefit seen for serum addition (*Figure 6*). This leads us to believe that L-serine functions as a molecular cue that directs *Salmonella* toward serum, but nutrients present in serum other than L-serine provide the major growth advantages.

## Structure of *S. enterica* Tsr in complex with L-serine

We next undertook structural studies to understand the specific recognition of L-serine by Tsr. The full-length Tsr protein includes a periplasmic ligand-binding domain (LBD), a transmembrane HAMP domain, and a cytosolic coiled-coil region, which oligomerizes to form trimers-of-dimers, and complexes with the downstream chemotaxis signaling components, CheA and CheW (*Figure 7A*; *Cassidy et al., 2020*). No experimentally determined structure had been published for *S. enterica* Tsr (*Se*Tsr) and the single experimentally determined Tsr structure to have captured the L-serine-binding interactions is a crystal structure of *Ec*Tsr LBD of moderate resolution (2.5 Å, PDB: 3ATP), in which the electron density for the ligand is weak and the orientation of the L-serine ligand is ambiguous (*Tajima et al., 2011*). Despite the poorly defined binding site interactions, this *Ec*Tsr crystal structure

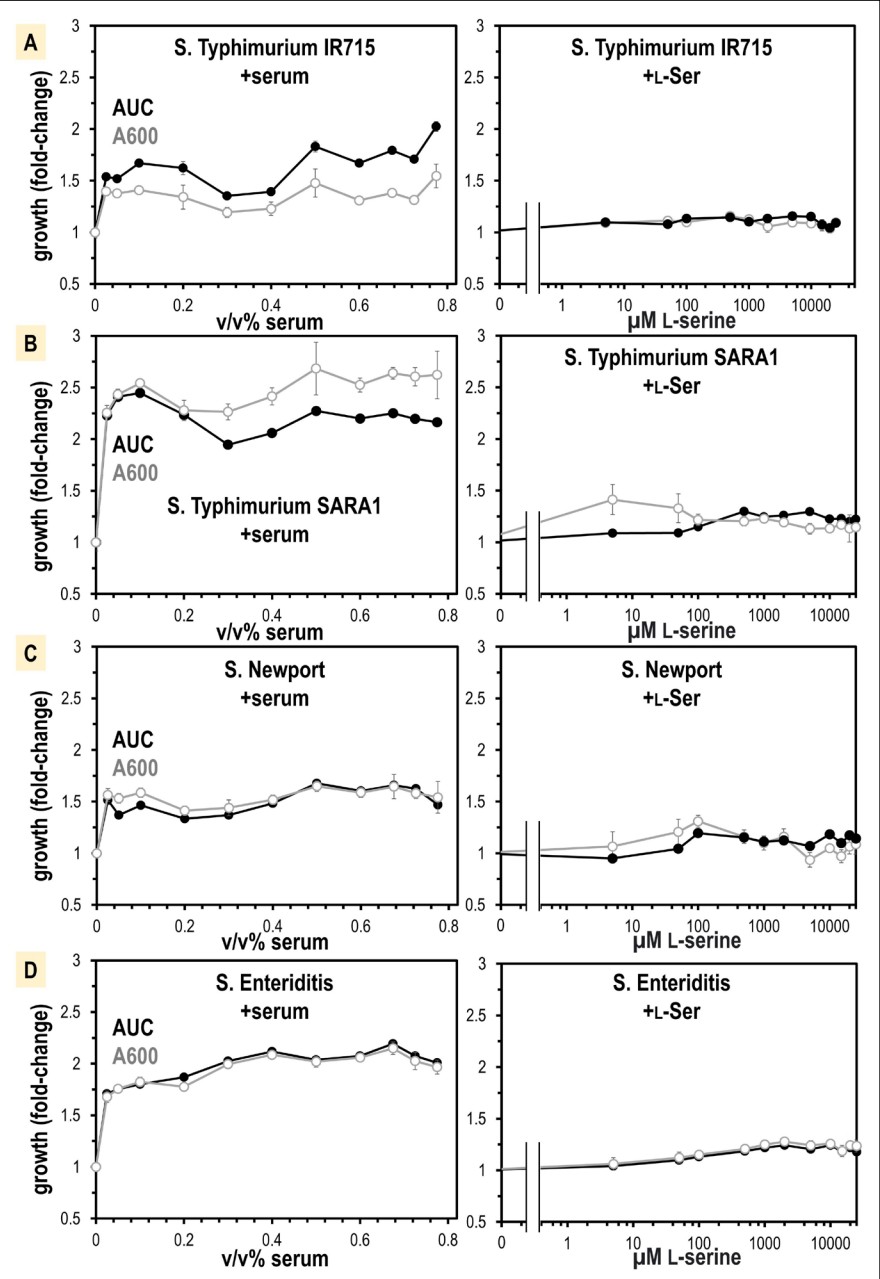

**Figure 6.** Non-typhoidal *S. enterica* strains obtain a growth benefit from human serum not recapitulated from L-Ser treatments alone. (**A–D**) Growth is shown as area under the curve (AUC, black) and A600 at mid-log phase for the untreated replicates (gray, n = 16). Data are means, and error is SEM.

The online version of this article includes the following figure supplement(s) for figure 6:

**Figure supplement 1.** Total serine present in human serum samples, as determined by mass spectrometry.

has guided numerous other studies of chemoreceptor signal transduction and nanoarray function (*Cassidy et al., 2020*; *Burt et al., 2020*).

We recognized that in the prior *Ec*Tsr study the methods described exchanging the protein crystal into a glycerol cryoprotectant. We hypothesized that during the glycerol soak serine leached out of the crystal leaving the binding site partially occupied and caused the electron density to be weak for the ligand and surrounding region. To capture a complex with a fully bound ligand site, we grew crystals of the soluble periplasmic portion of the *Se*Tsr LBD with L-serine, at a high salt concentration that would serve as a cryoprotectant without further manipulation, and harvested crystals directly from drops to

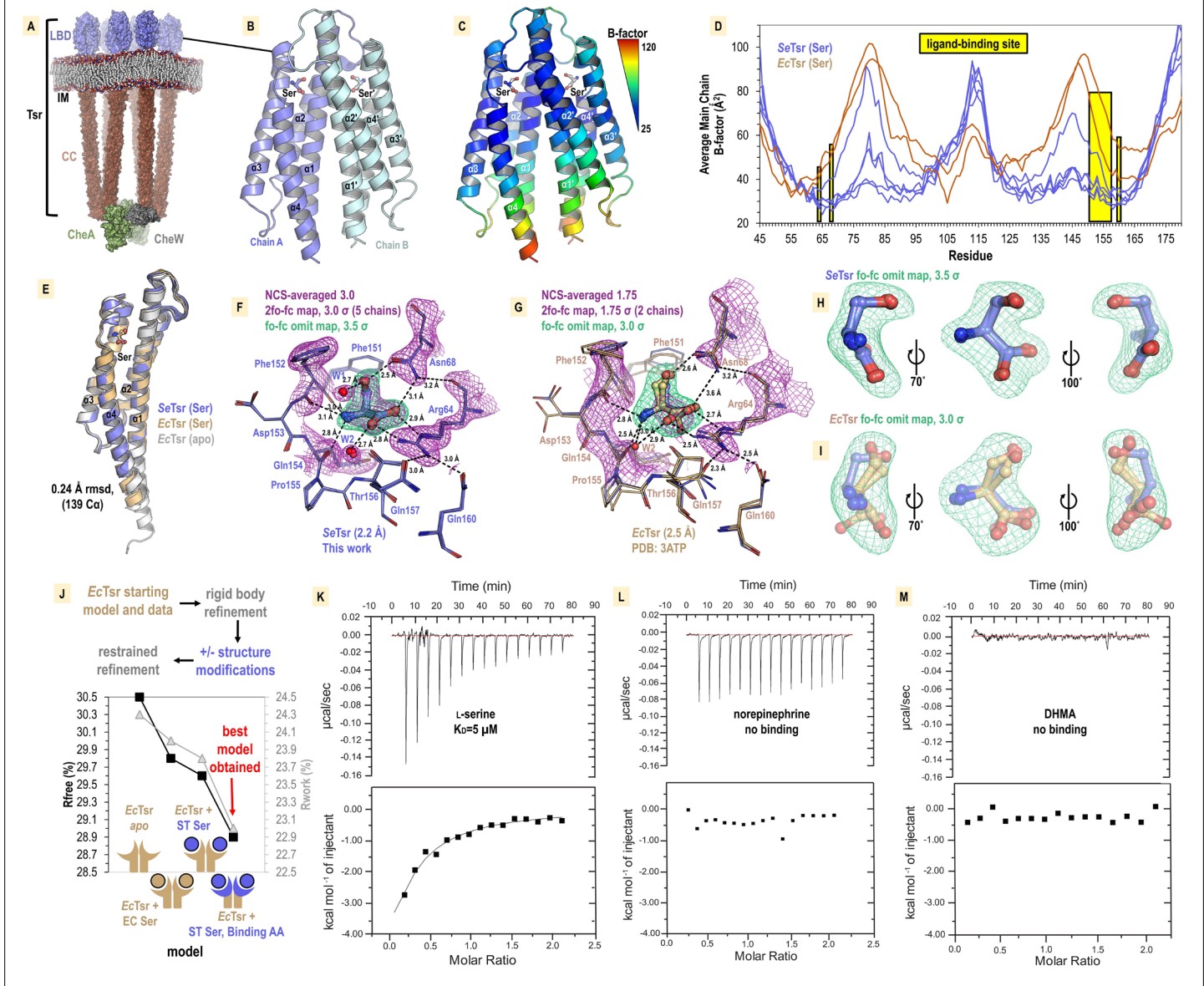

**Figure 7.** Structural mechanism underlying taxis to serum. (**A**) Model of the core chemoreceptor signaling unit showing two full-length Tsr chemoreceptor trimer-of-dimers; coiled-coil region (CC), inner membrane (IM) *Cassidy et al., 2020*. (**B**) Crystal structure of *S. enterica* Tsr ligand-binding domain (LBD) dimer in complex with L-serine (2.2 Å). (**C, D**) Relative order of the *Se*Tsr structure as indicated by B-factor (Å2). (**E**) Overlay of chains from serine-bound *Se*Tsr (blue), serine-bound *Ec*Tsr (orange), and apo *Ec*Tsr (white). (**F**) Binding of the L-serine ligand as seen with an overlay of the five unique chains of the asymmetric unit (AU) in the *Se*Tsr structure. Purple mesh represents averaged non-crystallographic symmetry 2f$_o$-f$_c$ omit map electron density (ligand not included in the density calculations). Green mesh represents f$_o$-f$_c$ omit map difference density for Chain A. Hydrogen bonds to the ligand are shown as dashed black lines with distances indicated in Å. (**G**) The ligand-binding site of serine-bound *Ec*Tsr is shown as in (**F**), with omit map f$_o$-f$_c$ electron density. The two chains of *Ec*Tsr in the AU are overlaid (orange) with one chain of serine-bound *Se*Tsr (blue). (**H, I**) Closeup view of the L-serine ligand and f$_o$-f$_c$ omit map density for the *Se*Tsr (blue) and *Ec*Tsr (orange) structures, respectively. (**J**) Paired refinement comparing resulting crystallographic R-factors, as indicated. The schematics indicate for each refinement which parts of the Tsr LBD dimer were modeled as in the *Ec*Tsr structure (orange) or *Se*Tsr structure (blue), with the crescent shape representing the ligand-binding site. For each refinement strategy and resulting model, R$_{free}$ values (black) and R$_{work}$ values (gray) are indicated. (**K–M**) Isothermal titration calorimetry analyses of the *Se*Tsr LBD with L-serine, norepinephrine (NE) , or 3,4-dihydroxymandelic acid (DHMA).

The online version of this article includes the following figure supplement(s) for figure 7:

**Figure supplement 1.** Comparison of *Se*Tsr ligand-binding domain (LBD) structures solved at different pH.

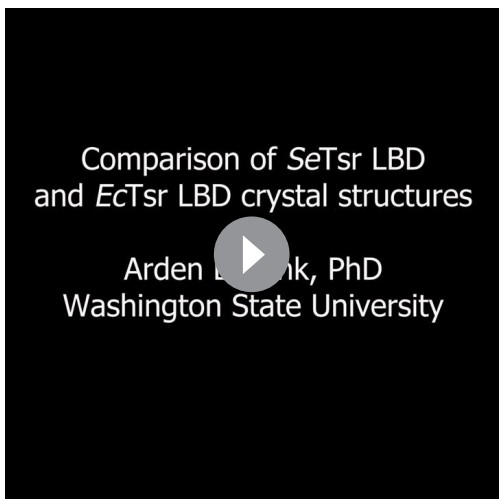

**Video 7.** *Se*Tsr ligand-binding domain (LBD) crystal structure (PDB: 8FYV) and comparison with *Ec*Tsr LBD (PDB: 3ATP). Crystal structure of *S. enterica* Typhimurium Tsr LBD (PDB: 8FYV) and comparison with *E. coli* Tsr LBD crystal structure (PDB: 3ATP). Viewable at https://youtu.be/OlowDhRLNhA. https://elifesciences.org/articles/93178/figures#video7

prevent leaching of the ligand from the binding site. After conducting extensive crystallization trials and examining X-ray diffraction data from over 100 crystals, we identified a small number of *Se*Tsr LBD crystals with sufficient quality for structure determination. The crystal that provided the highest quality data (2.2 Å resolution, PDB: 8FYV) was grown in a mixture of buffers of pH 7.5–9.7 (refer to 'Methods', *Supplementary file 1*). To assess whether the basic pH had any impact on the protein structure, we solved a second structure from a crystal grown at pH 7–7.5 (*Supplementary file 1*, *Figure 7—figure supplement 1*, PDB: 8VL8). By overlaying the two structures, it is apparent they are nearly identical, with no observable changes at the ligand-binding site (*Figure 7—figure supplement 1*). Because the structure obtained at higher pH (PDB: 8FYV) exhibits superior quality and clearer electron density in critical regions of interest, it is this structure we refer to in subsequent sections.

The crystal structure of *Se*Tsr LBD contains five monomers in the asymmetric unit, providing five independent views of the L-serine binding site, with homodimers formed between chains A and B, C and D, and E and its crystal symmetry mate, E' (*Figure 7B*). Lower B-factors in the *Se*Tsr ligand binding region are indicative of greater order, reflecting that our structure is fully serine-bound (*Figure 7C and D*). *Se*Tsr and *Ec*Tsr possess high-sequence similarity in the LBD region—100% identity of the ligand-binding residues and 82.1% identity over the entire periplasmic domain—and as expected, retain a similar global structure with all chains of serine-bound *Se*Tsr (5), serine-bound *Ec*Tsr (2), and apo *Ec*Tsr (2) overlaying within 0.24 Å rmsd over 139 $C_\alpha$ (*Figure 7E*).

## Molecular recognition of L-serine by Tsr

The higher resolution (+0.3 Å) of the *Se*Tsr structure, and full occupancy of the ligand, provides a much-improved view of the interactions that facilitate specific recognition of L-serine (*Figure 7F*). Using non-crystallographic symmetry averaging, we leveraged the five independent *Se*Tsr monomers to generate a well-defined $2f_o$-$f_c$ map of the L-serine ligand and residues involved in ligand coordination (*Figure 7F*, *Video 7*). Omit-map difference density, which is calculated in the absence of a modeled ligand and reduces potential bias in the electron density map, was fit well by the placement of the L-serine ligand (*Figure 7F*). In this orientation, the ligand is in an optimal energetic conformation that satisfies all possible hydrogen bonding interactions: the positively charged peptide amine donates hydrogen bonds to the backbone carbonyl oxygens of Phe151, Phe152, and Gln154, the negatively charged ligand carboxyl group donates hydrogen bonds to the Arg64 guanidinium group, the Asn68 side change amine, and a water (W2), and the ligand hydroxyl sidechain donates a hydrogen bond to the Asn68 sidechain oxygen, and accepts a hydrogen bond from a water (W1) (*Figure 7F*, *Video 7*). All five chains of the *Se*Tsr structure are consistent in the positions of the ligand and surrounding residues (*Figure 7F*, *Video 7*).

With the aid of the improved view provided by our *Se*Tsr structure, we noticed the L-serine is positioned differently than what was modeled into the weak density of the *Ec*Tsr structure (*Figure 7G–I*, *Video 7*). The *Ec*Tsr structure has the serine positioned with the sidechain hydroxyl facing into the pocket toward Asn68, and the orientations of Asn68, Phe152, Asp153, and Gln157 of the ligand binding pocket are modeled inconsistently, without justification, between the two *Ec*Tsr chains in the asymmetric unit (*Figure 7G–I*). Calculating $f_o$-$f_c$ omit map density for both structures shows not only that our new *Se*Tsr ligand orientation fits the density of our structure well, but that it is a better fit

for the data from the *Ec*Tsr structure (*Figure 7H and I*). This is apparent by how the *Ec*Tsr serine $C_\beta$ and sidechain hydroxyl are misaligned with the curvature of the $f_o$-$f_c$ map and that the carboxylate is partially outside of the electron density (*Figure 7I*).

An analysis that can quantifiably determine which positions of the serine and ligand-binding residues result in the best model is to perform pairwise refinements and compare the crystallographic $R_{work}$ and $R_{free}$ statistics (*Figure 7J*). Using as our starting model the *Ec*Tsr homodimer, and its deposited data from the protein databank, we performed refinements using identical strategies with three scenarios: no serine modeled in the binding site (*apo*), the published *Ec*Tsr serine pose, or the serine pose from our *Se*Tsr structure (*Figure 7J*). The poorest resulting *Ec*Tsr model was *apo*, yielding $R_{work}$/$R_{free}$ values of +0.3%/+0.7% relative to the deposited model. This result supports the presence of the serine ligand in the structure. Substituting the serine with the pose observed in the *Se*Tsr structure led to an improved model, reducing the $R_{work}$/$R_{free}$ values by –0.2%/–0.2%. Lastly, we replaced both the serine ligand and the ligand-binding residues of the *Ec*Tsr model with those from our *Se*Tsr structure, which resulted in the highest quality model with meaningfully lower $R_{work}$/$R_{free}$ values of –1.0%/–0.9% (*Figure 7J*). Consequently, we can conclude that the correct serine pose and ligand-binding site positions for both models are those we determined using the higher resolution *Se*Tsr structure, in which the L-serine side chain hydroxyl faces outward from the pocket, toward the solvent, and every hydrogen bonding group of the ligand is satisfied by the residues of the binding site (*Figure 7F and J*).

## SeTsr LBD binds L-serine, but not NE or DHMA

Despite a lack of chemotactic responses to NE and DHMA in our CIRA experiments (*Figure 4—figure supplement 1*, *Videos 5 and 6*, https://public.vetmed.wsu.edu/Baylink), our uncertainty lingered as to whether these neurotransmitters are sensed through Tsr. Prior work created theoretical models of NE and DHMA binding *Ec*Tsr at the L-serine site and proposed this as the molecular mechanism underlying *E. coli* chemoattraction to these compounds (*Orr et al., 2020*). However, with our new experimentally determined *Se*Tsr LBD crystal structure in hand, it is clear that the hydrogen bonding network is specific for L-serine (*Figure 7F*, *Video 7*), and we were doubtful such dissimilar ligands as NE or DHMA would be accommodated. We note that the amino acids that constitute the L-serine binding pocket are identical between *Se*Tsr and *Ec*Tsr (*Figure 7F and G*), and so we reason that *Se*Tsr LBD and *Ec*Tsr LBD should have similar molecular functions and ligand specificity.

Despite several studies reporting a direct role of Tsr in directing chemoattractant responses to NE and DHMA, no direct evidence of ligand-binding has ever been presented. Thus, we performed isothermal titration calorimetry (ITC) experiments, a gold standard for measuring protein–ligand interactions, to study ligand recognition by *Se*Tsr LBD (*Figure 7K–M*). As expected, L-serine produces a robust exothermic binding curve, exhibiting a $K_D$ of approximately 5 μM (*Figure 7K*). This matches well with our observation of *S. enterica* chemoattraction requiring a source of L-serine at least 0.5–5 μM (*Figure 5D and E*) and with prior ITC data with *Ec*Tsr LBD, which also reported a 5 μM $K_D$ (*Tajima et al., 2011*). Conversely, neither NE nor DHMA showed any evidence of binding to *Se*Tsr LBD (*Figure 7L and M*). These data confirm that *Se*Tsr is a receptor for L-serine, and not NE or DHMA, and further support the molecular mechanism of Tsr-mediated serum attraction to be through L-serine.

## Relevance of bacterial taxis to serum in other systems

Our chemotaxis experiments with the *tsr* strain show that while Tsr is not the only chemoreceptor involved in taxis to serum for *S. enterica*, it does play an important role through chemoattraction to the high concentration of L-serine within serum. We hypothesized that attraction to serum, mediated through Tsr, might extend beyond *S. enterica* to other bacterial species. The biological distribution of Tsr chemoreceptors has not been previously characterized, so we applied our structural insights to define a motif consisting of the amino acids involved in L-serine recognition. When we analyzed the genomes of Enterobacteriaceae genera such as *Escherichia*, *Citrobacter*, and *Enterobacter,* we found their Tsr orthologues to possess this motif (*Figure 8A*). We next conducted a comprehensive search for chemoreceptor genes containing the L-serine recognition motif, creating a database containing all organisms that possess putative Tsr orthologues (*Figure 8B*, *Supplementary file 2*). The biological distribution of Tsr reveals the chemoreceptor to be widely conserved among Gammaproteobacteria, particularly within the families Enterobacteriaceae, Morganelliaceae, and Yersiniaceae (*Figure 8B*).

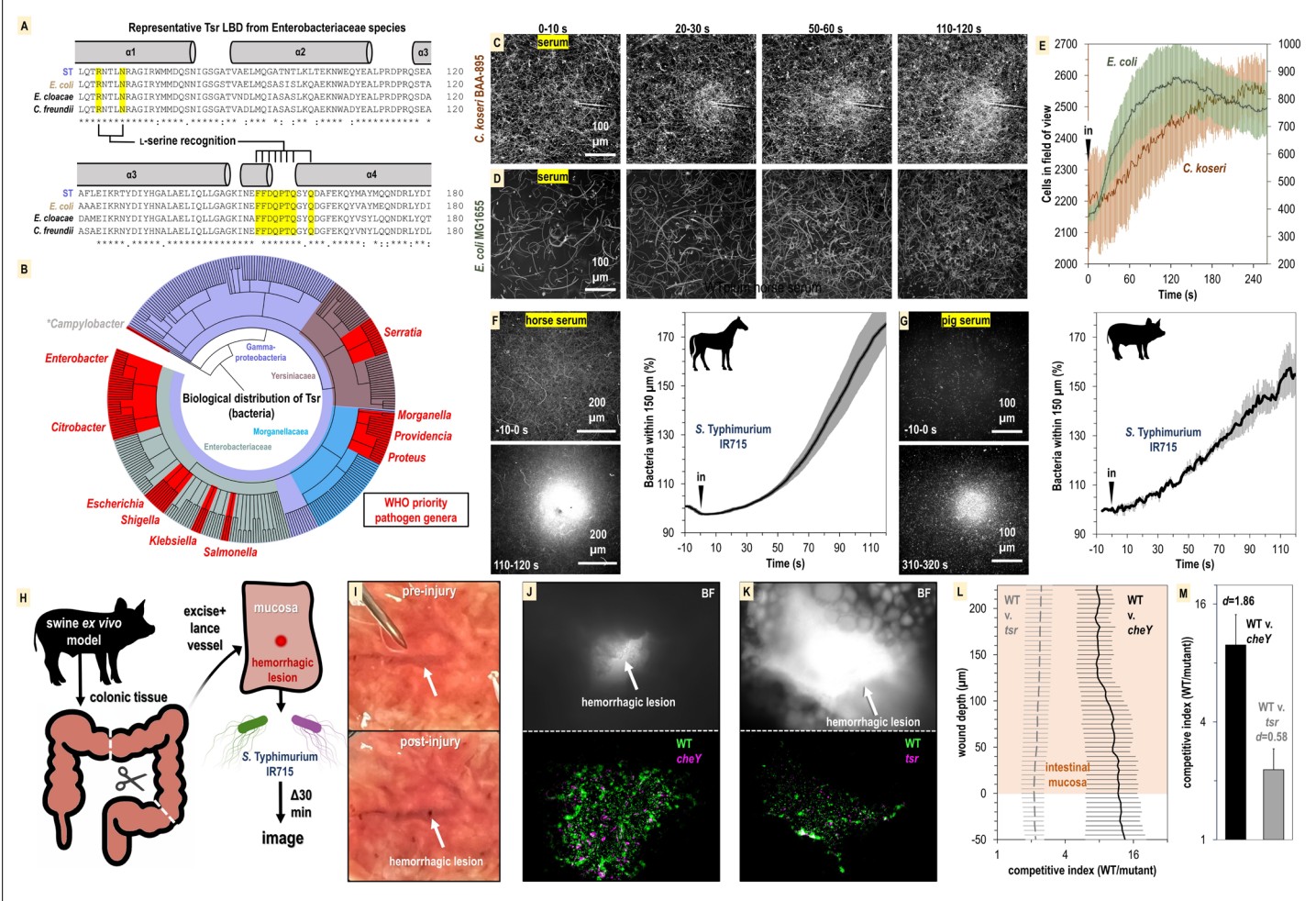

**Figure 8.** Relevance of bacterial vampirism in other systems. (**A**) Sequence conservation of the Tsr ligand-binding domain among representative Enterobacteriaceae orthologues with residues of the L-serine ligand-binding pocket highlighted. (**B**) Biological distribution of Tsr orthologues. WHO priority pathogen genera are highlighted in red. (**C, D**) Response of *C. koseri* BAA-895 (n = 3, 37°C) and *E. coli* MG1655 to human serum (n = 3, 30°C), shown as max projections at indicated time points. (**E**) Quantification of chemotaxis responses. Plotted data shown are shown on separate y-axes as mean cells in the field of view averaged over 1 s. (**F, G**) Response of wildtype (WT) *S*. Typhimurium IR715 to horse serum or pig serum. (**H**) Overview of swine ex vivo colonic hemorrhagic lesion model. (**I**) Representative images of colonic mucosa before (top) and after (bottom) lancing of vasculature. (**J, K**) Representative images showing bacterial localization into colonic hemorrhagic lesions in co-inoculation experiments with WT (*gfp*) and *cheY* (*dtom*) (n = 4), or WT (*gfp*) and *tsr* (*dtom*) (n = 3) *S*. Typhimurium IR715. The lesion is shown in brightfield (BF, top) and gfp/dtom fluorescent channels (bottom). (**L**) Competitive indices of migration into lesions. The mucosa is set to a y-axis value of 0, and migration further into the lesion is reflected by increasing y-axis values. The x-axis is shown on a log scale. (**M**) Area under the curve (AUC) quantification of bacterial lesions localization shown in (**L**) (0–225 μM). Effect size (Cohen's *d*) is noted. Data shown are means, and error bars show SEM.

We discovered that many WHO priority pathogens are among the species that have Tsr, leading us to suspect these pathogens and pathobionts also perform serum taxis.

To address this question, we performed CIRA analyses with human serum and the bacterial strains *E. coli* MG1655 and *C. koseri* BAA-895, the latter being a clinical isolate with previously uncharacterized chemotactic behavior. We find that both these strains exhibit attraction to human serum on time scales and magnitudes similar to experiments with *Salmonella* (*Figure 8C–E*, *Videos 8 and 9*). Without further genetic analyses in these strain backgrounds, the evidence for Tsr mediating serum taxis for these bacteria remains circumstantial. Nevertheless, taxis to serum appears to be a behavior shared by diverse Enterobacteriaceae species and perhaps also Gammaproteobacteria priority pathogen genera that possess Tsr such as *Serratia*, *Providencia*, *Morganella*, and *Proteus* (*Figure 8B*).

So far, our focus has been on bacterial taxis toward human serum. However, given the presence of L-serine in the serum of various animal species, we were interested in exploring if serum from

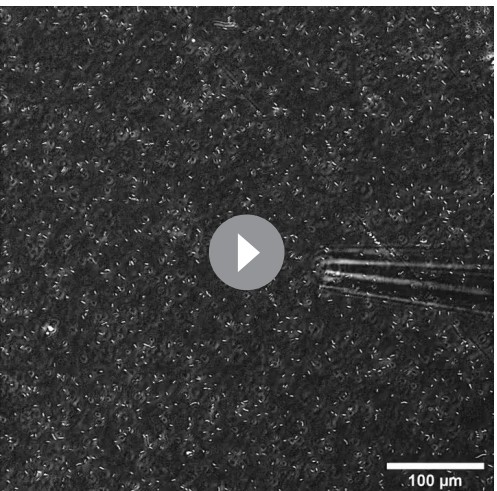

**Video 8.** Representative chemosensory injection rig assay (CIRA) experiment showing response of *C. koseri* BAA-895 to human serum. Viewable at https://youtu.be/9iMJz2OPbso.

https://elifesciences.org/articles/93178/figures#video8

other mammals could also attract bacteria. We prepared horse serum from whole horse blood through clotting and centrifugation to use in CIRA experiments with *S.* Typhimurium IR715. Similar to human serum, the bacterial population rapidly swam toward the horse serum source, with the relative number of cells in close proximity to the source nearly doubling within 120 s of exposure (*Figure 8F*). To further investigate this concept, we utilized serum we collected from a third species, a single domestic pig. The data revealed that pig serum similarly induces a rapid attraction of *S.* Typhimurium (*Figure 8G*). This suggests that the phenomenon of serum attraction is relevant to diverse host-microbe systems and is not specific to bacteria associated with humans. Considering the data from all CIRA experiments, while some differences are observed in strain responses to serum sources (*S. enterica* serovars, *E. coli*, and *C. koseri*) and serum from different animals (human, horse, and pig), all WT strains examined share the behavior of rapid taxis toward serum.

As a preliminary investigation into whether serum taxis is involved in bacterial pathogenesis, we developed an enterohemorrhagic lesion model to investigate if chemotaxis and Tsr can mediate bloodstream entry. Because swine share many aspects of human gastrointestinal physiology (*Meurens et al., 2012*), and we established that swine serum stimulates pathogen attraction (*Figure 8G*), we utilized swine colonic tissue from the same animal for ex vivo experimentation (*Figure 8H*). We created an enterohemorrhagic lesion by a single lance of the tissue vasculature from the mucosal side, and then exposed the wound to a 1:1 mixture of motile *S.* Typhimurium IR715 WT and chemotaxis mutant (*Figure 8H and I*, https://public.vetmed.wsu.edu/Baylink). Following 30 min of exposure, the localization of the bacteria in proximity to the lesion was determined through fluorescence microscopy and enumerated as a competitive index of WT vs. mutant. By imaging through the z-plane in 5-micron intervals we were also able to quantify bacterial colonization as a function of wound depth. In the context of this ex vivo model, we observed that the bacteria were readily able to penetrate >200 µm into the wounded vasculature (*Figure 8J–M*). Strikingly, the WT efficiently localizes toward, and migrates into, the hemorrhagic lesion, outcompeting the *cheY* and *tsr* mutants by approximately 10:1 and 2:1 (*Figure 8J–M*). These results align with our other data indicating that serum taxis requires CheY, and that Tsr contributes, but is not the only chemoreceptor involved in mediating attraction. Additional investigation is required to confirm the occurrence of these behaviors during infection, but it seems plausible that bacterial taxis toward sources of serum in the host gut environment occurs, potentially contributing to the development of bacteremia.

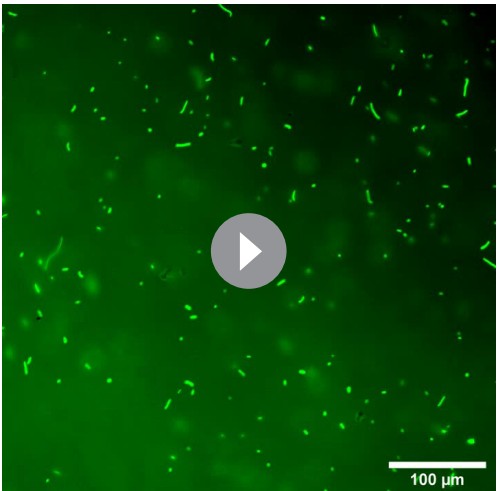

**Video 9.** Representative chemosensory injection rig assay (CIRA) experiment showing response of *E. coli* (pxS-gfp) MG1655 to human serum. Viewable at https://www.youtube.com/watch?v=jq3cj9e52n4.

https://elifesciences.org/articles/93178/figures#video9

## Discussion

The most common outcome for gastrointestinal bacterial infections is that they are resolved by the immune system without chronic infection or lingering pathologies (*Figure 1*). To understand

why, in rare circumstances, infections do persist, become chronic, systemic, and life-threatening, we must uncover the mechanisms at the crossroads where pathogens divert from the routine course of infection and do something unusual. An emerging area of interest in the axis of gut-microbe health is how the behavior of gut microbes can shift in response to inflammation, antibiotic usage, and pathogen invasion, into a state of dysbiosis recalcitrant to treatment (*Belizário and Faintuch, 2018*; *Rivera-Chávez et al., 2017*; *Zeng et al., 2017*; *Kitamoto et al., 2020*). We can learn about the factors that stimulate microbial-induced pathologies by studying how bacteria respond to host-derived stimuli that are unique to the diseased gut (*Kitamoto et al., 2020*). Here, we have investigated how bacterial chemosensing functions to respond to a source of serum, a chemical feature enteric bacteria only encounter in the event of GI bleeding.

We demonstrated that serum from humans and other animals contains components that act as chemoattractants and nutrients for Enterobacteriaceae known as instigators of enteric bleeding and causal agents of bacteremia and sepsis (*Figures 2, 3, 6, and 8*). Bacterial infiltration of wounded enteric vasculature, mediated by chemotaxis and the chemoreceptor Tsr, suggests serum attraction plays a role in the bloodstream entry of these bacterial species. In a broader context, the attraction of Enterobacteriaceae to serum aligns with an emerging understanding of how bacterial chemotaxis can drive tropism for sites of damaged tissue, injury, and inflammation (*Figure 1*; *Zhou et al., 2023*; *Scales and Huffnagle, 2013*; *Aihara et al., 2014*; *Callahan et al., 2021*; *Croxen and Finlay, 2010*; *Hanyu et al., 2019*; *Rivera-Chávez et al., 2013*). This phenomenon of bacterial attraction to serum through chemotaxis to access serum nutrients represents a novel pathogenesis strategy, which we term 'bacterial vampirism' (*Figure 9*).

## A new ecological context for L-serine chemoattraction

We show here that the bacterial attraction response to serum is robust and rapid; the motile population reorganizes within 1–2 min of serum exposure to be significantly biased toward the serum source (*Figures 2 and 8*). Serum taxis occurs through the cooperative action of multiple bacterial chemoreceptors that perceive several chemoattractant stimuli within serum, one of these being the chemoreceptor Tsr through recognition of L-serine (*Figure 4*). While Tsr is known as a bacterial sensor of L-serine (*Zhou et al., 2023*; *Keegstra et al., 2022*; *Cassidy et al., 2020*), the physiological sources of L-serine in the gut sensed by Tsr have not been well-defined. Although a major source is dietary, recent research has suggested that L-serine from damaged tissue may play a significant role in the diseased gut environment and drive opportunistic pathogenesis. In a study using a dextran sodium sulfate (DSS)-colitis model to investigate L-serine utilization by Enterobacteriaceae, L-serine was identified as a critical nutrient providing metabolic and growth advantages in the inflamed gut (*Kitamoto et al., 2020*). Moreover, colitis was found to stimulate substantial increases in the luminal availability of most amino acids, with the average serine content nearly doubling (*Kitamoto et al., 2020*). A notable characteristic of the DSS-colitis model is intestinal bleeding (*Kim et al., 2012*), suggesting that the

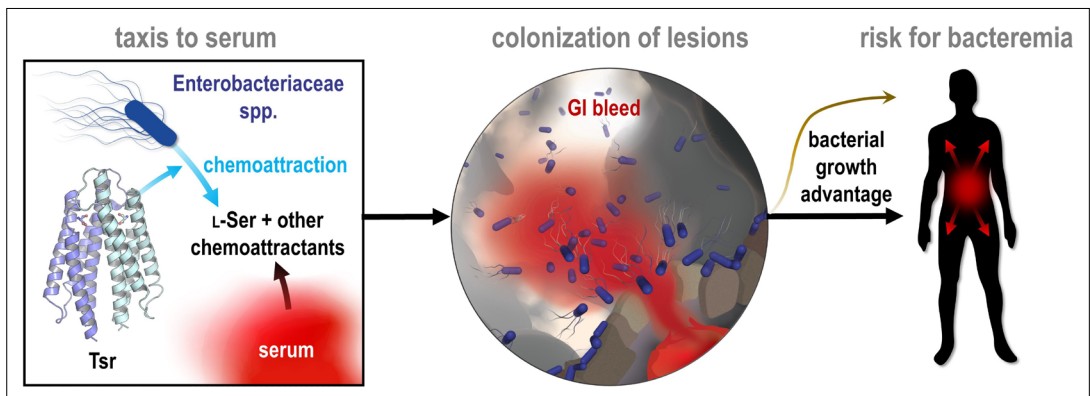

**Figure 9.** Model of bacterial vampirism. Serum contains high concentrations of L-Ser, and other chemoattractants, that are recognized by chemoreceptors, including Tsr, to drive Enterobacteriaceae taxis toward serum. Taxis to serum promotes colonization of enterohemorrhagic lesions and provides a bacterial growth advantage from nutrients acquired from serum. The bacterial behavior of seeking and feeding on serum, which we have defined as 'bacterial vampirism' may represent a risk factor for bacterial entry into the bloodstream.

enrichment of amino acids, including serine, could be attributed to the leakage of host metabolites from serum. The ability of pathogenic Enterobacteriaceae to exploit growth advantages from L-serine during colitis may be linked to their highly sensitive and rapid chemoattraction to this effector emitted from human serum.

Our structural and molecular investigations into *S. enterica* Tsr affirm its role as a serine sensor, revealing a highly optimized and selective ligand-binding site for the recognition of L-serine (*Figure 7*, *Video 7*). Although the host-derived metabolites NE and DHMA have been reported to be chemo-attractants for *E. coli* sensed directly through Tsr, we were unable to corroborate these results in *S.* Typhimurium in chemotaxis experiments, nor binding assays with the purified recombinant LBD, despite the high-sequence conservation between the two orthologues (*Figure 4—figure supplement 1*, *Figures 7L and M and 8A*, https://public.vetmed.wsu.edu/Baylink). Our inability to substantiate a structure–function relationship for NE/DHMA signaling indicates these neurotransmitters are not ligands of Tsr, or, at the very least, are not broadly relevant as chemoeffectors sensed through Tsr orthologues. In contrast, we saw that L-serine drove chemoattraction to a source as low as 500 nM, all strains tested were attracted to L-Ser, a binding $K_D$ to *Se*Tsr LBD of 5 µM was established through ITC, and was captured in complex in the *Se*Tsr LBD crystal structure with clearly defined electron density (*Figures 5 and 7*). Our study is not the only one to raise questions regarding the physiological relevance of NE/DHMA as chemoattractants. A study examining *E. coli* sensing of NE/DHMA by Tsr showed little chemoattraction occurs in the range of 10–100 µM, and at higher (millimolar) concentrations a chemorepulsion response was observed (*Lopes and Sourjik, 2018*). Notably, the concentrations required in the latter study to observe a chemotactic response are far above what is thought to be physiologically relevant for tissue or serum (*Tong et al., 1996*). Given that treatment of serum with serine racemase diminishes taxis toward serum and eliminates the WT taxis advantage over *tsr* (*Figure 4—figure supplement 1*), our data support that L-serine is the chemoattractant sensed through Tsr that confers serum attraction.

An unanticipated result from obtaining the first structures of *Se*Tsr LBD (PDB: 8FYV) is that the well-resolved ligand-binding site contradicts key interactions as they were modeled in the earlier *Ec*Tsr LBD structure (3ATP), apparently owing to the difficulty of interpreting the weak electron density (*Figure 7F–I*). Through comparative refinements, we show that changing the ligand position and binding site residues of the *Ec*Tsr LBD model to be as they are in the *Se*Tsr LBD structure demonstrably improves the model (*Figure 7J*). Therefore, the differences in the models are not attributable to them being two different proteins, rather the *Ec*Tsr LBD structure is modeled incorrectly, and both structures should be modeled with the serine pose and ligand-binding site as they are in the *Se*Tsr LBD structure (*Figure 7F*). The *Ec*Tsr LBD structure has informed the creation of full-length atomic models of Tsr, and the core signaling unit, for use in molecular simulations and fitting of cryo-EM data (*Cassidy et al., 2020*; *Burt et al., 2020*; *Cassidy et al., 2015*). Since we now have in hand a higher resolution and better-resolved Tsr crystal structure, future studies of this sort should benefit from the improved understanding of L-serine recognition. The *Se*Tsr LBD structure was also helpful in accurately defining the signature motif of Tsr orthologues, which allowed us to mine genomic databases and report the first extensive characterization of which bacterial species possess this chemoreceptor (*Figure 8A and B*). From this database of Tsr sequences, we were able to visualize the distribution of Tsr orthologues across biology, which strikingly are enriched among Enterobacteriaceae, but are also present in several other families that include WHO priority pathogens (*Figure 8B*, *Supplementary file 2*). This analysis confirms that the bacterial species most commonly associated with bacteremia in patients with IBD possess the Tsr chemoreceptor (*Figure 8B*, *Supplementary file 2*; *Goren et al., 2020*).

In the context of *Salmonella* pathogenesis, it is interesting to note that diverse serovars, which vary in terms of host specificity and epidemiology, exhibit serum attraction and have the ability to utilize serum as a nutrient (*Figures 3 and 6*). The specific nutrients responsible for the growth benefits derived from serum remain undefined; however, one potential candidate is the presence of energy-rich glycoconjugates, which were previously implicated as nutrients for bacteria within the inflamed intestine based on the upregulation of galactose utilization operons and the prevalence of lectin-positive stained tissue (*Stecher et al., 2008*). Enteric Peyer's patches are primary invasion sites for non-typhoidal *Salmonella*, and these structures are situated close to vasculature that can be damaged through the pathogen's destruction of microfold (M) cells, causing localized bleeding (*Shi et al., 2012*). *S. enterica* Typhimurium uses the chemoreceptor Tsr to locate and invade Peyer's patches

(*Rivera-Chávez et al., 2016*), which could involve L-serine chemoattraction originating from serum or necrotic cells. Although GI bleeding is relatively uncommon in *Salmonella* infections overall, it afflicts 60% of infected children under the age of five (*White et al., 2019*). Therefore, while we acknowledge that serum attraction is not a routine pathogenesis strategy employed by *Salmonella*, the significant number of bacterial-induced GI bleeding cases and the association between GI bleeding and bacterial invasion into the bloodstream provide opportunities for bacterial vampirism to be involved in infection outcomes (*Figure 9*; *Weber et al., 2020*; *Vohra et al., 2020*; *Irving et al., 2021*).

# Materials and methods

**Key resources table**

| Reagent type (species) or resource | Designation | Source or reference | Identifiers | Additional information |
|---|---|---|---|---|
| Strain, strain background (*Salmonella enterica* Typhimurium) | *Salmonella enterica* Typhimurium | *Rogers et al., 2020* | IR715 | Nalidixic acid-derivative of ATCC 14028 |
| Strain, strain background (*S. enterica* Typhimurium) | *S. enterica* Typhimurium *cheY* mutant | *Rogers et al., 2020* | FR13 | IR715 *cheY*::Tn*10* (Tet^R) |
| Strain, strain background (*S. enterica* Typhimurium) | *S. enterica* Typhimurium *tsr* mutant | *Rogers et al., 2020* | FR4 | IR715 *tsr*::pFR3 (Cm^R) |
| Strain, strain background (*S. enterica* Typhimurium) | *S. enterica* Typhimurium Clinical Isolate | *Beltran et al., 1991* | SARA1 | Isolated from patient in Mexico |
| Strain, strain background (*S. enterica* Newport) | *S. enterica* Newport Clinical Isolate | *Shariat et al., 2013b* | M11018046001A | Isolated from patient in PA, USA |
| Strain, strain background (*S. enterica* Enteriditis) | *S. enterica* Enteriditis Clinical Isolate | *Shariat et al., 2013a* | 05E01375 | Isolated from patient in PA, USA |
| Strain, strain background (*Citrobacter koseri*) | *C. koseri* Clinical Isolate | ATCC | BAA-895 | Human Clinical Isolate |
| Strain, strain background (*Escherichia coli*) | *E. coli* Clinical Isolate | Karen Guillemin (UO) Millipore Sigma | MG1655, K12 | Model of WT *E. coli* Rosetta (DE3) BL21 derivative |
| Biological sample (*Homo sapiens*) | Human serum | This study | Prod. #: ISERABOTCHI100ML | See method details |
| Recombinant DNA reagent | XS Plasmid expressing sfGFP | *Wiles et al., 2018* | pXS-sfGFP | pGEN-mcs with a modular sfGFP expression scaffold (Amp^R) |
| Recombinant DNA reagent | XS Plasmid expressing mPulm | *Wiles et al., 2018* | pXS-mPlum | pGEN-mcs with a modular mPlum expression scaffold (Amp^R) |
| Recombinant DNA reagent | pET-30a(+)-*Se*TsrLBD | This study; Genscript | | Vector for recombinant expression of SeTsr LBD (Kan^R) |
| Peptide, recombinant protein | *Se*Tsr LBD | This study | | See 'Method details' |

## Lead contact

Further information and requests for resources and reagents should be directed to and will be fulfilled by the lead contact, Arden Baylink (arden.baylink@wsu.edu).

## Materials availability

Strains and plasmids generated in this study will be made available upon request by the Lead Contact with a completed Materials Transfer Agreement.

## Experimental model and subject details

### Bacterial strains

Strains of *S. enterica*, *E. coli*, and *C. koseri* used in this study are described in the Key resources table. To generate fluorescent strains, electrically competent bacterial cultures were prepared through successive washing of cells with ice-cold 10% glycerol, and then transformed by electroporation (Bio-Rad GenePulser Xcell) with pxS vectors containing either sfGFP, or mPlum, and AmpR genes (*Wiles*

*et al., 2018*). Transformants were isolated through growth on selective media and stored as glycerol stocks for subsequent use. To prepare motile bacterial cells for CIRA experiments, bacterial cultures were grown shaking overnight in 2–5 ml tryptone broth (TB) and 50 µg/ml ampicillin (TB+Amp) at 30°C or 37°C, as needed. The following day, 25 µl of overnight culture was used to inoculated 25 ml of fresh TB+Amp and grown shaking for 3–5 hr at the desired temperature to reach $A_{600}$ of approximately 0.5. To better isolate responses to effectors of interest and remove confounding variables, such as pH variation and quorum-signaling molecules, we washed and exchanged the cells into a buffer of defined composition. Bacterial cultures were centrifuged at 1500 × *g* for 20 min and exchanged into a chemotaxis buffer (CB) containing 10 mM potassium phosphate (pH 7), 10 mM sodium lactate, and 100 µM EDTA. Cultures were diluted to approximately $A_{600}$ 0.2 (or as indicated in figure legends) and allowed to recover rocking gently for 30–60 min to become fully motile.

For in vitro growth analyses, *S. enterica* strains were grown in Luria-Bertani (LB) media shaking overnight at 37°C. The following day, cultures were pelleted by centrifugation and resuspended in a minimal media (MM) containing 47 mM $Na_2HPO_4$, 22 mM $KH_2PO_4$, 8 mM NaCl, 2 mM $MgSO_4$, 0.4% glucose (w/v) 11.35 mM $(NH_4)_2SO_4$, 100 µM $CaCl_2$. 5 µl of the overnight cultures at 0.05 $A_{600}$ were used to inoculate fresh solutions of 200 µl of MM, or additives diluted into MM (human serum or L-serine), in a 96-well microtiter plate. A plate reader was used to monitor cell growth while shaking at 37° via $A_{600}$ readings every 5 min.

## Method details

### Chemosensory injection rig assay (CIRA)

Our CIRA system is based on prior work, with several notable changes to methodology, as described (*Huang et al., 2015*; *Perkins et al., 2019*; *Howitt et al., 2011*). The CIRA apparatus was constructed using a pump for injection (either a refurbished Eppendorf Transjector 5246 or Femtojet 4i) and universal capillary holder, with solutions injected through Femtotip II glass microcapillaries (Eppendorf), and the microcapillary position controlled with an MP-285 micromanipulator (Sutter) at a 30° angle of attack. To generate a microgradient, a constant flow from the microcapillary was induced by applying compensation pressure ($P_c$) of 35 hPa, unless otherwise specified. The stability of the microgradient under these conditions was determined with Alexa488 dye (Thermo Fisher), and the flow was determined empirically with methylene blue dye to be approximately 300 fl/min at 30°C (*Figure 2—figure supplement 1*). Pooled off-the-clot human serum was obtained from Innovative Research; human serum was deidentified and had no additional chemical additives and is presumed to be fully active. Horse serum was prepared from whole horse blood from Innovative Research. Pig serum was prepared from whole blood collected from a single animal. Treatment solutions were filtered through a 0.2 µm filter and injected as-is (serum) or diluted in CB (serine, aspartate, NE, DHMA). Serine racemase treatment of human serum was performed with the addition of 5 µl of proprietary serine racemase solution from a DL-serine assay kit (Abcam) to 1 ml of serum for 3 hr. For each CIRA experiment, a fresh pond of 50 µl of motile bacteria was mounted open to air on a 10-well slide (MP Biomedicals), and the microcapillary containing the treatment of interest was lowered into the pond. Bacterial responses were imaged with an inverted Nikon Ti2 Eclipse microscope with an enclosed heated sample chamber. Temperature of experiments were at 37°C, unless otherwise indicated.

### CIRA microgradient modeling

Diffusion is modeled as a 3D process in which the diffusive species is slowly and continuously introduced at a fixed point in a large ambient fluid volume. The species to be injected is prepared at concentration $M_s$ (typically in the range of 0.5 µM to 5 mM) and is injected at a volume rate Q = 305.5 fl/min. The species diffuses into the ambient fluid with diffusion constant $D$. The governing equation for the diffusion of a species introduced continuously at a point source is:

$$C(r, t) = \frac{q}{4\pi D r} erfc \frac{r}{2\sqrt{Dt}}$$

where $r$ is the distance from the point source, $t$ is the time from injection initiation, and $q = M_s Q$ is the injection rate of the species, and $C$ is the species concentration. We can simplify the presentation by defining a characteristic length scale $r_0$, characteristic time $t_0$, and dimensionless variables as:

$$r_0 = \frac{Q}{4\pi D}, \; t_0 = \frac{Q^2}{(8\pi)^2 D^3}, \; \bar{r} = \frac{r}{r_0}, \; \bar{t} = \frac{t}{t_0}.$$

Then, we have the diffusion-driven concentration model:

$$C\left(\bar{r}, \bar{t}\right) = \frac{M_s}{\bar{r}} \, erfc \, \frac{\bar{r}}{\sqrt{\bar{t}}}$$

Representative diffusion coefficients are A488, $4.00 \times 10^{-6} \, \mathrm{cm}^2/\mathrm{s}$ ; L-serine, $8.71 \times 10^{-6} \, \mathrm{cm}^2/\mathrm{s}$ ; L-aspartate, $9.35 \times 10^{-6} \, \mathrm{cm}^2/\mathrm{s}$ . Due to the small injection rate, our assumption of a point source leads to a model that is valid at distances $r \gg r_0 \sim 1 \, \mathrm{nm}$ and times $t \gg t_0 \sim 1 \, \mathrm{ns}$. We also consider the total species quantity integrated along a viewing direction. The result is:

$$I\left(\bar{r}, \bar{t}\right) = 2M_s r_0 \int_0^{\infty} \frac{d\bar{z}}{\sqrt{\bar{z}^2 + \bar{r}^2}} \, erfc \, \frac{\sqrt{\bar{z}^2 + \bar{r}^2}}{\bar{t}}$$

where $\bar{z}$ is the unitless integration variable along the viewing direction. This integral was evaluated numerically by considering the sequence of points $z_k = \frac{k+1}{2}\Delta\bar{z}$ for $k = 1, 2, 3, \ldots$ and $\Delta\bar{z}$ some small interval. Then, we have the discrete integral approximation:

$$I\left(\bar{r}, \bar{z}\right) \approx 2M_s r_0 \Delta\bar{z} \sum_{k=1}^{N} \left(z_k^2 + \bar{r}^2\right)^{-1/2} erfc \, \frac{\sqrt{z_k^2 + \bar{r}^2}}{\bar{t}}$$

Computations for this work used 1 μm steps extending out to $r = 500 \, \mathrm{m}$. That is, $\Delta\bar{z} = 1 \, \mathrm{m}/r_0$ and $N = 500$.

## Ex vivo swine hemorrhagic lesion model

Swine intestinal tissue was procured from a single 8-week-old domestic farm-reared Yorkshire cross-breed animal (the same animal from which serum was drawn), in accordance with Washington State University Institutional Animal Care and Use Committee and Institutional Biosafety Committee approval. Intact colonic tissue was excised, and an incision was made along its length to flatten the tissue and expose the mucosa. Sections of tissue approximately 2.5 cm² in size were prepared for experimentation through gentle washing with saline solution, followed by CB, to remove fecal contents and debris. A sterile Gawal Sharp Point was used to lance the vasculature to generate a model enterohemorrhagic lesion. The lesion site was positioned in a MatTek dish over a pond of 50 μl of motile bacterial cells (1:1 fluorescently tagged WT and mutant in CB, at a total $A_{600}$ of 1.0). After 30 min, bacterial localization into the lesion was visualized using an inverted Nikon Ti2 Eclipse microscope. Images were captured at ×20 magnification proceeding through the Z-plane in 5 μm intervals.

## Cloning and recombinant protein expression

Cloning of the *S. enterica* Tsr LBD construct for recombinant protein expression was performed as a service by Genscript Biotech Corp. The sequence of the periplasmic portion of the ligand-binding domain of *S. enterica* Typhimurium Tsr (gene *STM4533*), corresponding to residues 32–187 of the full-length protein, was encoded into a pet-30a(+) vector (Tsr-LBD-pet-30a(+)), at the NdeI and HindIII sites, with a short N-terminal TEV cleavage tag (MENLYFQ) such that the final expressed protein sequence was:

MENLYFQSLKNDKENFTVLQTIRQQQSALNATWVELLQTRNTLNRAGIRWMMDQSNIGSGATVA
ELMQGATNTLKLTEKNWEQYEALPRDPRQSEAAFLEIKRTYDIYHGALAELIQLLGAGKINEFF
DQPTQSYQDAFEKQYMAYMQQNDRLYDIAVEDNNS

Chemically competent Rosetta BL21(DE3) *E. coli* (MilliporeSigma) were transformed by heat shock with the Tsr-LBD-pet-30a(+) vector, and transformants identified by growth on selective media containing 20 μg/ml (LB + Kan). Cells were grown overnight in 5 ml LB + Kan. The following day, 1 ml of overnight culture was used to inoculate 1 l of fresh LB + Kan, and cultures were grown to $OD_{600}$ 0.6–0.8 and induced with 0.4 mM isopropyl β-D-1-thiogalactopyranoside (IPTG). After growth for 3 hr at 37°C, cells were harvested by centrifugation.

## Purification of recombinant SeTsr LBD

The cell pellet was resuspended in a lysis buffer containing 50 mM Tris pH 7.5, 0.1 mM DTT, 1 mM EDTA, 5 mg DNAse I, and 1 cOmplete protease inhibitor tablet per 1 l of culture (Sigma-Aldrich), and cells were lysed by sonication. Afterward, the lysate was kept on ice and adjusted to 20% ammonium sulfate saturation and stirred at 4°C for 30 min. The lysate was centrifuged at 15,000 rpm for 30 min in a Beckman ultracentrifuge. The soluble fraction was retained, and an ammonium precipitation trial was conducted; the 20–40% fraction contained the majority of the Tsr LBD protein and was used for subsequent purification. The protein solution was dialyzed for 16 hr against 4 l of 20 mM Tris, pH 7.5, 20 mM NaCl, and 0.1 mM EDTA, then run over an anion exchange column and FPLC (Akta). The purest fractions were pooled and treated with 0.3 mg/ml TEV protease, and the protein solution was dialyzed against 4 l of 50 mM Tris pH 8, 0.5 mM EDTA, and 1 mM EDTA for 48 hr at 4°C. Subsequently, the cleaved protein solution was exchanged into a buffer of 50 mM Tris pH 7.5, 1 mM EDTA, and 150 mM NaCl, and purified by gel filtration with an S200 column and FPLC. Pure protein fractions were pooled, concentrated to 7 mg/ml, and flash frozen in liquid $N_2$.

## Protein crystallography

Initial crystallization trials of $Se$Tsr LBD were performed with either TEV-cleaved or uncleaved protein samples at 7 mg/ml with ±2 mM L-serine and using 96-well matrix screens set up with a Mosquito robot (SPT Labtech). We only observed crystal hits with the cleaved, serine-treated crystals, the best of which was 0.056 M sodium phosphate, 1.344 M potassium phosphate, pH 8.2. This condition was further optimized to be 1.5 µl $Se$Tsr LBD protein (7 mg/ml), 0.5 µl of 8 mM L-serine, and 1.5 µl 1.69 M potassium phosphate pH 9.7, grown via hanging drop vapor diffusion with a 1 ml reservoir of 1.69 M potassium phosphate pH 9.7 at 22°C. Crystals were scooped directly from drops and flash frozen in liquid $N_2$. X-ray diffraction data were collected at the Berkeley Advanced Light Source (ALS) beamline 5.0.3. Out of over 100 crystals examined, only one diffracted to high resolution and was not impacted by crystal twinning. Data were indexed with DIALS (*Winter et al., 2022*), scaled with Aimless (*Winn et al., 2011*), and found to correspond well to space group C2$_1$. A conservative final resolution cutoff of 2.2 Å was applied on the basis of $CC_{1/2}$ >0.3 and completeness >50% in the highest resolution shell (*Karplus and Diederichs, 2012*).

The serine-bound $Ec$Tsr structure (PDB: 3ATP) was utilized as a molecular replacement search model with Phaser-MR in Phenix (*Adams et al., 2010*) to solve the $Se$Tsr LBD dataset with five monomers in the asymmetric unit. 10% of the data were designated as $R_{free}$ flags and the initial model was adjusted by setting all B-factors to 30 Å$^2$, and coordinates were randomized by 0.05 Å to reduce bias from the starting model. Subsequent model building with Coot (*Emsley and Cowtan, 2004*) and refinement with Phenix enabled placement of residues 42–182 and the serine ligand. However, residues 32–41 and 183–187 could not be resolved and were not modeled, causing the R/R$_{free}$ values to be elevated for a model of this resolution. Riding hydrogen atoms and translation, libration, screw refinement was applied and reduced R factors. The strongest remaining difference peak is along a symmetry axis. The final model R/R$_{free}$ values were 24.1/25.8%, with a 99th percentile all-atom MolProbity clashscore, and a 100th percentile overall MolProbity score (*Davis et al., 2007*).

Because the protein was in a buffer at pH 7.5, but the crystallization solution was at 9.7, there was uncertainty about the true pH of the crystalline sample and how this might impact ligand binding interactions. As a control, we crystallized the protein in a mother liquor of 1.62 M potassium phosphate pH 7, with 0.5 µl of 3 mM L-serine, at 22°C. The solution of this structure revealed no remarkable differences in the ligand binding interactions or in the global structure (*Figure 7—figure supplement 1*). The data quality suffered from ice rings, resulting in maps with high noise, and so we restricted refinement to a conservative 2.5 Å cutoff. The crystallographic statistics for these structures are listed in *Supplementary file 1*. The pH 7.5–9.7 and pH 7–7.5 structures were deposited to the protein data bank as entries 8FYV and 8VL8, respectively. On the basis of the higher quality maps, and stronger and more clearly interpreted electron density, we recommend structure 8FYV for use in future structural studies of $Se$Tsr.

For comparisons of model quality between $Se$Tsr LBD (8FYV) with $Ec$Tsr (3ATP), we re-refined 3ATP with its deposited data using rigid body refinement to obtain a starting model for subsequent evaluation. Then, we performed a series of restrained refinements with Phenix using identical $R_{free}$ flags and identical strategy of five cycles of xyz reciprocal space, xyz real space, individual isotropic B-factor, adp

weight optimization, and stereochemistry weight optimization. For these refinements, we used the *Ec*Tsr LBD structure ± the following modifications: (1) the deposited L-Ser present, (2) the deposited L-Ser removed (*apo*), (3) the L-Ser replaced by the *Se*Tsr L-Ser ligand, or (4) both the L-Ser and the ligand binding site residues replaced by those from the *Se*Tsr LBD structure. These comparisons show the best model, as evidenced by meaningfully reduced $R_{work}$ and $R_{free}$ values, is obtained with the position of the L-Ser and ligand binding residues from the higher resolved *Se*Tsr LBD structure (*Figure 7J*).

## ITC ligand binding studies

ITC experiments were performed in 50 mM Tris, 150 mM NaCl, 1 mM EDTA, pH 7.5, at 25°C. We dialyzed the protein into the experimental buffer and dissolved the small-molecule ligands into the same buffer. Protein concentrations were ~50 µM; titrant concentrations were 500 µM. Samples were degassed prior to experiments. All experiments were performed on a MicroCal ITC-200 system (GE Healthcare), with the gain set to 'low' and a syringe stir speed of 750 rpm. Titration data for the serine experiments were fit to a single-site binding model using the built-in ITC analysis software.

## Mass spectrometry

Determination of molar content of total serine in human serum samples was performed as a service through the University of Washington Mass Spectrometry Center. Samples were analyzed on the Waters TQ #1 instrument using a Thermo Hypersil Gold PFP column (2.1 × 100) with 0.1% heptafluorobutyric acid (HFBA) in water and acetonitrile.

## Quantification and statistical analysis

### Quantification of CIRA data

To determine the relative numbers of cells over time, a ratio of fluorescence intensity per cell was calculated using ImageJ. Fluorescence intensity was used as a readout of relative cell count over time using the 'plot profile' function in ImageJ (*Schneider et al., 2012*). Cell numbers were normalized to a baseline of '100%' at the start of treatment (shown as time 0). Distribution of the bacterial population was quantified through use of the 'radial profile' ImageJ plugin. Radial distribution data were normalized by setting the field of view periphery as the baseline of 'onefold', which we defined as 240 µm distance from the source. Images and videos shown were processed using the 'enhance contrast' function in ImageJ and adjusting intensity thresholds to normalize fluorescence intensity per cell across channels. For experiments with non-fluorescent cells, equivalent procedures were performed using phase-contrast data and enumeration of cells over time using a MATLAB-based tracking software (*Perkins et al., 2019*).

## Statistical analyses

Data from replicate experiments were averaged and interpreted on the basis of their mean, standard error of the mean, and effect sizes. Effect sizes for data are indicated as Cohen's *d* value:

$$d = \frac{M_1 - M_2}{\sigma_{pooled}}$$

where:

$$\sigma_{pooled} = \sqrt{\frac{\sigma_1^2 + \sigma_2^2}{2}}$$

Where relevant, p-values were calculated using either one-sided or unpaired two-sided tests, with significance determined at p<0.05:

$$t = \frac{\bar{x}_1 - \bar{x}_2}{\sqrt{s^2 \left(\frac{1}{n_1} + \frac{1}{n_2}\right)}}$$

where

$$s^2 = \frac{\sum\limits_{i=1}^{n_1} \left(x_i - \bar{x}_1\right)^2 + \sum\limits_{j=1}^{n_2} \left(x_j - \bar{x}_2\right)^2}{n_1 + n_2 - 2}$$

## Acknowledgements

Funding for this work was provided by NIAID under award numbers 1K99AI148587 and 4R00AI148587-03, and startup funding from Washington State University to AB. We thank Karen Guillemin (University of Oregon) for the Eppendorf Transjector used for the CIRA experiments. We thank Nikki Shariat (University of Georgia, Athens), Nkuchia Mikanatha and Pennsylvania NARMS and GenomeTrakr Programs, and Andreas Bäumler (University of California, Davis) for providing the *Salmonella* strains used in this work. Beamline 5.0.3 of the Advanced Light Source, a DOE Office of Science User Facility under Contract No. DE-AC02-05CH11231, is supported in part by the ALS-ENABLE program funded by the National Institutes of Health, National Institute of General Medical Sciences, grant P30 GM124169-01. All research on human and animal samples was performed in accordance with, and approval of, the Institutional Biosafety Committee and Institutional Animal Care and Use Committee at Washington State University.

## Additional information

### Competing interests

Arden Baylink: owns Amethyst Antimicrobials, LLC. The other authors declare that no competing interests exist.

### Funding

| Funder | Grant reference number | Author |
| --- | --- | --- |
| National Institute of Allergy and Infectious Diseases | 1K99AI148587 | Arden Baylink |
| National Institute of Allergy and Infectious Diseases | 4R00AI148587-03 | Arden Baylink |
| Washington State University | Startup fund | Arden Baylink |

The funders had no role in study design, data collection and interpretation, or the decision to submit the work for publication.

### Author contributions

Siena J Glenn, Data curation, Formal analysis, Investigation, Methodology, Writing – original draft, Writing – review and editing; Zealon Gentry-Lear, Data curation, Investigation, Methodology, Writing – review and editing; Michael Shavlik, Data curation, Formal analysis, Investigation, Writing – review and editing; Michael J Harms, Data curation, Formal analysis, Supervision, Investigation, Writing – review and editing; Thomas J Asaki, Software, Formal analysis, Investigation, Writing – review and editing; Arden Baylink, Conceptualization, Resources, Data curation, Formal analysis, Supervision, Funding acquisition, Validation, Investigation, Visualization, Methodology, Writing – original draft, Project administration, Writing – review and editing

### Author ORCIDs

Siena J Glenn https://orcid.org/0000-0001-6762-1264
Zealon Gentry-Lear https://orcid.org/0000-0002-0350-7494
Michael J Harms https://orcid.org/0000-0002-0241-4122
Arden Baylink https://orcid.org/0000-0001-5522-769X

### Ethics

This study was performed in strict accordance with the recommendations in the Guide for the Care and Use of Laboratory Animals of the National Institutes of Health. All of the animals were handled

according to approved institutional animal care and use committee (IACUC) protocols (#7128) of the Washington State University. The protocol was approved by the Institutional Biosafety Committee (IBC) at Washington State University (#1372).

Reviewer #1 (Public review): https://doi.org/10.7554/eLife.93178.3.sa1
Reviewer #3 (Public review): https://doi.org/10.7554/eLife.93178.3.sa2
Author response https://doi.org/10.7554/eLife.93178.3.sa3

---

## Additional files

### Supplementary files
Supplementary file 1. Table of crystallographic statistics.
Supplementary file 2. Database of Tsr homologue sequences.
MDAR checklist

### Data availability
Diffraction data have been deposited in PDB under the accession code 8FYV and 8VL8. Source data for growth curves, lesion model, CIRA with L-Ser, DHMA, and NE can be downloaded at https://public.vetmed.wsu.edu/Baylink. Experiment conditions are noted within 'README.txt' files. The high-resolution microscopy videos are >20TB in size so it is not practical to make these data available via a public repository; however these data are available upon request from the corresponding author. Any of the data presented in our work would be shared with the requesters, rapidly and without restriction.

The following datasets were generated:

| Author(s) | Year | Dataset title | Dataset URL | Database and Identifier |
|---|---|---|---|---|
| Baylink A, Gentry-Lear Z, Glenn S | 2023 | *Salmonella enterica* serovar Typhimurium chemoreceptor Tsr (taxis to serine and repellents) ligand-binding domain in complex with l-serine | https://www.rcsb.org/structure/8FYV | RCSB Protein Data Bank, 8FYV |
| Glenn S, Baylink A | 2024 | *Salmonella enterica* Typhimurium taxis to serine and repellents (Tsr) ligand-binding domain with L-Ser, pH 7 | https://www.rcsb.org/structure/8VL8 | RCSB Protein Data Bank, 8VL8 |

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
