## [Editor Report · eLife assessment]

This work uses an interdisciplinary approach combining microfluidics, structural biology, and genetic analyses to provide **important** findings that show that pathogenic enteric bacteria exhibit taxis toward human serum. The data are **compelling** and show that the behavior utilizes the bacterial chemotaxis system and the chemoreceptor Tsr, which senses the amino acid L-serine. The work provides an ecological context for the role of serine as a bacterial chemoattractant and could have clinical implications for bacterial bloodstream invasion during episodes of gastrointestinal bleeding.

---

## [Referee Report · Reviewer #1 (Public review)]

Updated summary:

Glenn et al. present solid evidence that both lab and clinical *Salmonella enterica* serovars rapidly migrate towards human serum using an exciting approach that combines microfluidics, structural biology and genotypic analysis. The authors succeed in bringing to light a novel context for the role of serine as a bacterial chemoattractant as well as documenting what is likely to be a key step in bloodstream entry for some of the main sepsis-associated pathogens during gastrointestinal bleeding. They illustrate the generality of their findings through phylogenetic analysis, testing additional species within the Enterobacteriaceae family and showing attraction towards swine and equine serum. Their interdisciplinary approach here greatly increases the scope of their findings.

I would also like to note that, whilst I enjoyed the interdisciplinary scope of this study, I am personally not well placed to review the protein structural aspects of this work.

Additional strengths of the revised manuscript:

All weaknesses raised in my review of the original manuscript have been satisfactorily addressed in the revised manuscript. It is interesting to note that the accumulation pattern of the bacteria 50-75 um from the source of serum could, as the author's now note, be due to the avoidance of bactericidal serum elements. Alternative explanations, however, could include chemoreceptor saturation (i.e. close to the serum source, high ligand concentrations could saturate chemoreceptors preventing further chemotaxis) or Weber's Law considerations (cell's ability to detect a given change in chemical concentrations diminishes with increasing background concentrations - thus, as cells get closer to the serum source, their ability to chemotax decreases).

The authors have also added new experimental data and analyses and these constitute major new strengths of the revised manuscript:

- The authors show that the competitive advantage of WT cells relative to a tsr mutant is removed when serum is treated with serine-racemase and this provides strong evidence that chemotaxis towards serine is responsible for the reduced attraction of the tsr mutant towards serum (i.e. rather than any possible pleiotropic effects).

- New experimental data showing *Salmonella enterica* is also attracted to swine and equine serum (including an ex vivo swine model) is a useful addition that hints at the potential generality of the response reported here.

- The authors now include additional data to back up the intriguing lack of a movement response towards norepinephrine and DHMA reported here.

Additional weaknesses of the revised manuscript:

- The addition of an ex vivo swine model is an exciting new inclusion in the updated manuscript. However, information regarding biological and technical replication here is currently unclear or missing.

---

## [Referee Report · Reviewer #3 (Public review)]

Summary:

This manuscript characterizes a chemoattractant response to human serum by pathogenic bacteria, focusing on pathogenic stratins of *Salmonella enterica* Se. The researchers conduct the chemotaxis assays using a micropipette injection method that allows real-time tracking of bacterial population densities. They found that clinical isolates of several Se strains present a chemoattractant response to human serum. The specific chemoattractant within the serum is identified as L-serine, a highly characterized and ubiquitous chemoattractant, that is sensed by the Tsr receptor. They further show that chemoattraction to serum is impaired with a mutant strain devoid of Tsr. X-ray crystallography is then used to determine the structure of L-serine in the Se Tsr ligand binding domain, which differs slightly from a previously determine structure of a homologous domain. They went on to identify other pathogens that have a Tsr domain through a bioinformatics approach and show that these identified species also present a chemoattractant response to serum.

Strengths and Weaknesses:

This study is well executed and the experiments are clearly presented. These novel chemotaxis assays provide advantages in terms of temporal resolution and ability to detect responses from small concentrations. That said, it is perhaps not surprising these bacteria respond to serum as it is known to contain high levels of known chemoattractants, serine certainly, but also aspartate. In fact, the bacteria are shown to respond to aspartate and the tsr mutant is still chemotactic. The authors do not adequately support their decision to focus exclusively on the Tsr receptor. Tsr is one of the chemoreceptors responsible for observed attraction to serum, but perhaps, not the receptor. Furthermore, the verification of chemotaxis to serum is a useful finding, but the work does not establish the physiological relevance of the behavior or associate it with any type of disease progression. I would expect that a majority of chemotactic bacteria would be attracted to it under some conditions. Hence the impact of this finding on the chemotaxis or medical fields is uncertain.

The authors also state that "Our inability to substantiate a structure-function relationship for NE/DHMA signaling indicates these neurotransmitters are not ligands of Tsr." Both norepinephrine (NE) and DHMA have been shown previously by other groups to be strong chemoattractants for *E. coli* (Ec), and that this behavior was mediated by Tsr (e.g. single residue changes in the Tsr binding pocket block the response). Given the 82% sequence identity between the Se and Ec Tsr, this finding is unexpected (and potentially quite interesting). To validate this contradictory result the authors should test *E. coli* chemotaxis to DHMA in their assay. It may be possible that Ec responds to NE and DHMA and Se doesn't. However, currently the data is not strong enough to rule out Tsr as a receptor to these ligands in all cases. At the very least the supporting data for Tsr being a receptor for NE/DHMA needs to be discussed.

The authors also determine a crystal structure of the SeTsr periplasmic ligand binding domain bound to L-Ser and note that the orientation of the ligand is different than that modeled in a previously determined structure of lower resolution. I agree that the SeTsr ligand binding mode in the new structure is well-defined and unambiguous, but I think it is too strong to imply that the pose of the ligand in the previous structure is wrong. The two conformations are in fact quite similar to one another and the resolution of the older structure, is, in my view, insufficient to distinguish them. It is possible that there are real differences between the two structures. The domains do have different sequences and, moreover, the crystal forms, and cryo-cooling conditions are different in each case. It's become increasingly apparent that temperature, as manifested in differential cooling conditions here, can affect ligand binding modes. It's also notable that full-length MCPs show negative cooperativity in binding ligands, which is typically lost in the isolated periplasmic domains. Hence ligand binding is sensitive to the environment of a given domain. In short, the current data is not convincing enough to say that a previous "misconception" is being corrected.

---

## [Author Response]

The following is the authors’ response to the original reviews.

We appreciate the thoughtful review of our manuscript by the reviewers, along with their valuable suggestions for enhancing our work. In response to these suggestions, we conducted additional experiments and made significant revisions to both the text and figures. In the following sections, we first highlight the major changes made to the manuscript, and thereafter address each reviewer's comments point-by-point. We hope these additional data and revisions have improved the robustness and clarity of the study and manuscript. Please note that as part of a suggested revision we have changed the manuscript title to be: Bacterial vampirism mediated through taxis to serum.

Major revisions and new data:

(1) We conducted additional experiments testing taxis to serum using a swine ex vivo enterohemorrhagic lesion model in which we competed wildtype versus chemotaxis deficient strains (Fig. 8). We selected swine for these experiments due to their similarity in gastrointestinal physiology to humans. In these experiments we see that chemotaxis, and the chemoreceptor Tsr, mediate localization to, and migration into, the lesion. We also tested, and confirmed, taxis to serum from swine and serum from horse, that supporting that serum attraction is relevant in other host-pathogen systems.

(2) We present additional experimental data and quantification of chemotaxis responses to human serum treated with serine-racemase (Fig. S3). This treatment reduces wildtype chemoattraction and the wildtype no longer possesses an advantage over the tsr strain, providing further evidence that L-serine is the specific chemoattractant responsible for Tsr-mediated attraction to serum.

(3) We present additional data in the form of 17 videos of chemotaxis experiments with norepinephrine and DHMA showing null-responses under various conditions. These data provide additional support to the conclusion that these chemicals are not responsible for bacterial attraction to serum. We have included these raw data as a new supplementary file (Data S1) for those in the field that are interested in these chemicals.

(4) Based on comments from Reviewer 2 regarding whether the position of the ligand and ligand-binding site residues in the previously-reported EcTsr LBD structure are incorrect, or whether these differences are due to the proteins being from different organisms, we performed paired crystallographic refinements to determine which positions result in model improvement (Fig. 7J). Altering the EcTsr structure to have the ligand and ligandbinding site positions from our new higher resolution and better-resolved structure of *Salmonella* Typhimurium Tsr results in a demonstrably better model, with both Rwork and Rfree lower by about 1% (Fig. 7J). These data support our conclusion that the correct positions for both structures are as we have modeled them in the S. Typhimurium Tsr structure. We also solved an additional crystal structure of SeTsr LBD captured at neutral pH (7-7.5) that confirms our structure captured with elevated pH (7.5-9.7) has no major changes in structure or ligand-binding interactions (Fig. S6, Table S2).

(5) Based on comments from Reviewer 2 on the accuracy of the diffusion calculations, we present a new analysis (Fig. S2) comparing the experimentally-determined diffusion of A488 compared to its calculated diffusion. We found that:

[line 111]: “As a test case of the accuracy of the microgradient modeling, we compared our calculated values for A488 diffusion to the normalized fluorescence intensity at time 120 s. We determined the concentration to be accurate within 5% over the distance range 70270 µm (Fig. S2). At smaller distances (<70 µm) the measured concentration is approximately 10% lower than that predicted by the computation. This could be due to advection effects near the injection site that would tend to enhance the effective local diffusion rate.”

(6) Both reviewers asked us to better justify why we focused on the chemoreceptor Tsr, and had questions about why we did not investigate Tar. The low concentration of Asp in serum suggests Tar could have some effect, but less so than Trg or Tsr (see Fig. 4A). We have revised the text throughout to better convey that we agree multiple chemoreceptors are involved in the response and clarify our rationale for studying the role of Tsr:

[line 178]: “We modeled the local concentration profile of these effectors based on their typical concentrations in human serum (Fig. 4B). Of these, by far the two most prevalent chemoattractants in serum are glucose (5 mM) and L-serine (100-300 µM) (Fig. 4B-F). This suggested to us that the chemoreceptors Trg and/or Tsr could play important roles in serum attraction.”

[line 186]: “Since tsr mutation diminishes serum attraction but does not eliminate it, we conclude that multiple chemoattractant signals and chemoreceptors mediate taxis to serum. To further understand the mechanism of this behavior we chose to focus on Tsr as a representative chemoreceptor involved in the response, presuming that serum taxis involves one, or more, of the chemoattractants recognized by Tsr that is present in serum: L-serine, NE, or DHMA.”

[line 468] “Serum taxis occurs through the cooperative action of multiple bacterial chemoreceptors that perceive several chemoattractant stimuli within serum, one of these being the chemoreceptor Tsr through recognition of L-serine (Fig. 4).”

Point-by-point responses to reviewer comments:

**Reviewer #1:**
(1) Presumably in the stomach, any escaping serum will be removed/diluted/washed away quite promptly? This effect is not captured by the CIRA assay but perhaps it might be worth commenting on how this might influence the response in vivo. Perhaps this could explain why, even though the chemotaxis appears rapid and robust, cases of sepsis are thankfully relatively rare.

To clarify, the Enterobacteriaceae species we have tested here are colonizers of the intestines, not the stomach, and cases of bacteremia from these species are presumably due to bloodstream entry through intestinal lesions. Whether or not intestinal flow acts as a barrier to bloodstream entry is not something we test here, and so we have not commented on this idea in the manuscript. We do demonstrate that attraction to serum occurs within seconds-to-minutes of exposure. We expect that the major protective effects against sepsis are the host antibacterial factors in serum, which are well-described in other work. We have been careful to state throughout the text that we see attraction responses, and growth benefits, to serum that is diluted in an aqueous media, which is different than bacterial growth in 100% serum or in the bloodstream.

(2) The authors refer to human serum as a chemoattractant numerous times throughout the study (including in the title). As the authors acknowledge, human serum is a complex mixture and different components of it may act as chemoattractants, chemo-repellents (particularly those with bactericidal activities) or may elicit other changes in motility (e.g. chemokinesis). The authors present convincing evidence that cells are attracted to serine within human serum - which is already a well-known bacterial chemoattractant. Indeed, their ability to elucidate specific elements of serum that influence bacterial motility is a real strength of the study. However, human serum itself is not a chemoattractant and this claim should be re-phrased - bacteria migrate towards human serum, driven at least in part by chemotaxis towards serine.

Throughout the text we have changed these statements, including in the title, to either be ‘taxis to serum’ or ‘serum attraction.’ On the timescales we tested our data support that chemotaxis, not chemokineses or other forms of direction motility, is what drives rapid serum attraction, since a motile but non-chemotactic cheY mutant cannot localize to serum (Fig. 4). We present evidence of one of these chemotactic interactions (L-Ser).

(3) Linked to the previous point, several bacterial species (including *E. coli* - one of the bacterial species investigated here) are capable of osmotaxis (moving up or down gradients in osmolality). Whilst chemotaxis to serine is important here, could movement up the osmotic gradient generated by serum injection play a more general role? It could be interesting to measure the osmolality of the injected serum and test whether other solutions with similar osmolality elicit a similar migratory response. Another important control here would be to treat human serum with serine racemase and observe how this impacts bacterial migration.

As addressed above, we have added additional experiments of serum taxis treated with serine racemase showing competition between WT and cheY, and WT and tsr (Fig. S3). These data support a role for L-serine as a chemoattractant driving attraction to serum. The idea of osmotaxis is interesting, but outside the scope of this work since we focus on chemoattraction to L-serine as one of the mechanisms driving serum attraction, and have multiple lines of evidence to support that.

(4) The migratory response of *E. coli* looks striking when quantified (Fig. 6C) but is really unclear from looking at Panel B - it would be more convincing if an explanation was offered for why these images look so much less striking than analogous images for other species (E.g. Fig. 6A).

We agree that the *E. coli* taxis to serum response is less obvious. We have brightened those panels to hopefully make it clearer to interpret (more cells in field of view over time). Also, as stated in the y-axes of these plots, this quantification was performed by enumerating the number of cells in the field of view, and the Citrobacter and Escherichia responses are shown on separate y-axes (now Fig. 8C). As indicated, the experiments have different numbers of starting motile cells, which we presume accounts for the difference in attraction magnitude. When investigating diverse bacterial systems we found there to be differences in motility under the culturing and experimental conditions we employed, for multiple reasons, and so for these data we thought it best to report raw cell numbers rather data normalized to the starting number of bacteria, as we do elsewhere. In the specific case of these *E. coli* responding to serum, please view Supplementary Movie S3, which both clearly shows the attraction response and that the bacteria grew in a longer, semi-filamentous form that seem to impair their swimming speed.

(5) It is unclear why the fold-change in bacterial distribution shows an approximately Gaussian shape with a peak at a radial distance of between 50 -100 um from the source (see for example Fig. 2H). Initially, I thought that maybe this was due to the presence of the microcapillary needle at the source, but the CheY distribution looks completely flat (Fig. 3I). Is this an artifact of how the fold-change is being calculated? Certainly, it doesn't seem to support the authors' claim that cells increase in density to a point of saturation at the source. Furthermore, it also seems inappropriate to apply a linear fit to these non-linear distributions (as is done in Fig. 2H and in the many analogous figures throughout the manuscript).

We have revised the text to address this point, and removed the comment about cells increasing in density to a point of saturation: [Line 138] “We noted that in some experiments the population peak is 50-75 µm from the source, possibly due to a compromise between achieving proximity to nutrients in the serum and avoidance of bactericidal serum elements, but this behavior was not consistent across all experiments. Overall, our data show *S. enterica* serovars that cause disease in humans are exquisitely sensitive to human serum, responding to femtoliter quantities as an attractant, and that distinct reorganization at the population level occurs within minutes of exposure (Fig. 3, Movie 2).”

We can confirm that this is not an artifact of quantification. Please refer to the videos of these responses, which demonstrates this point (Movies 1-5).

(6) The authors present several experiments where strains/ serovars competed against each other in these chemotaxis assays. As mentioned, these are a real strength of the study - however, their utility is not always clear. These experiments are useful for studying the effects of competition between bacteria with different abilities to climb gradients.However, to meaningfully interpret these effects, it is first necessary to understand how the different bacteria climb gradients in monoculture. As such, it would be instructive to provide monoculture data alongside these co-culture competition experiments.

Thank you for this suggestion. We agree that the coculture experiments showing strains competing for the same source of effector give a different perspective than monoculture. These experiments allow us to confirm taxis deficiencies or advantages with greater sensitivity, and ensure that the bacteria in competition have experienced the same gradient. This type of competition experiment is often used in in vivo experimentation for the same advantages. We note that in the gut the bacteria are not in monoculture and chemotactic bacteria do have to compete against each other for access to nutrients. Repeating all of the experiments we present to show both the taxis responses in coculture and monoculture would be an extraordinary amount of work that we do not believe would meaningfully change the conclusions of this study.

(7) Linked to the above point, it would be especially instructive to test a tsr mutant's response in monoculture. Comparing the bottom row of Fig. 3G to Fig. 3I suggests that when in co-culture with a cheY mutant, the tsr mutant shows a higher fold-change in radial distribution than the WT strain. Fig. 4G shows that a tsr mutant can chemotaxis towards aspartate at a similar, but reduced rate to WT. This could imply that (like the trg mutant), a tsr mutant has a more general motility defect (e.g. a speed defect), which could explain why it loses out when in competition with the WT in gradients of human serum, but actually seems to migrate strongly to human serum when in co-culture with a cheY mutant. This should be resolved by studying the response of a tsr mutant in monoculture.

Addressed above.

(8) In Fig. 4, the response of the three clinical serovars to serine gradients appears stronger than the lab serovar, whilst in Fig. 1, the response to human serum gradients shows the opposite trend with the lab serovar apparently showing the strongest response. Can the authors offer a possible explanation for these slightly confusing trends?

We suspect this relates to the fact that pure L-serine is a chemoattractant, whereas treatment with serum exposes the bacteria both to chemoattractants and, likely, chemorepellents. Strains may navigate the landscape of these stimuli different for a variety of reasons that are not simple to tease apart. The final magnitude of change in bacterial localization depends on multiple factors including swimming speed, adaptation, sensitivity of chemoattraction, and cooperative signaling of the chemoreceptor nanoarray. Thus, we cannot state with certainty how and why these strains are different across all experiments, but we can state that they are attracted to both serum and L-serine.

(9) In Fig. S2, it seems important to present quantification of the effect of serine racemase and the reported lack of response to NE and DHMA - the single time-point images shown here are not easy to interpret.

As suggested, we present quantification of the serum racemase treated samples (now Fig. S3). To assist in the interpretation of this max projections Fig. S3 now noted the chemotactic response (chemoattraction for L-serine, null-response for NE/DHMA). Further, we revised the text to state: [line 209]: “We observed robust chemoattraction responses to L-serine, evident by the accumulation of cells toward the treatment source (Fig. S3E, Movie 4), but no response to NE or DHMA, with the cells remaining randomly distributed even after 5 minutes of exposure (Fig. S3F-I, Movie 5, Movie S1).”

(10) Importantly, the authors detail how they controlled for the effects of pH and fluid flow (Line 133-136). Did the authors carry out similar controls for the dual-species experiments where fluorescent imaging could have significantly heated the fluid droplet driving stronger flow forces?

Most of our microfluidics experiments were performed in a temperature-controlled chamber (see Methods). Since the strains in the coculture experiments experienced the same experimental conditions we have no evidence of fluorescence-imaginginduced temperature changes that have impacted whether or not the bacteria are attracted to serum or the effectors we investigated.

(11) The inference of the authors' genetic analysis combined with the migratory response of *E. coli* and C. koseri to human serum shown in Fig. 6 is that Tsr drives movement towards human serum across a range of Enterobacteriaceae species. The evidence for the importance of Tsr here is currently correlative - more causal evidence could be presented by either studying the response of tsr mutants in these two species (certainly these should be readily available for *E. coli*) or by studying the response of these two species to serine gradients.

We have revised the text to state: [line 402] “Without further genetic analyses in these strain backgrounds, the evidence for Tsr mediating serum taxis for these bacteria remains circumstantial. Nevertheless, taxis to serum appears to be a behavior shared by diverse Enterobacteriaceae species and perhaps also Gammaproteobacteria priority pathogen genera that possess Tsr such as Serratia, Providencia, Morganella, and Proteus (Fig. 8B).”

We note that other work has thoroughly investigated *E. coli* serine taxis.

Figure Suggestions(1) Fig. 2 - The inset bar charts in panels H-J and the font size in their axes labels are too small - this suggestion also applies to all analogous figures throughout the manuscript.

We have increased the size of the text for these inset plots. We have also broken up some of the larger figures.

(2) Panel 2F - the cartoon bacterial cell and 'number of bacteria' are confusing and seem to contradict the y-axis label. This also applies to several other figures throughout the manuscript where the significance of this cartoon cell is quite hard to interpret.

As suggested, we have removed this cartoon.

(3) Panels G-I in Fig. 3 are currently tricky to interpret - it would be easier if the authors were to use three different colours for the three different strains shown across these panels.

We have broken up Figure 2 (which also had these types of plots) so that hopefully these labels are more clear. For the Figure in question (now Fig. 4), due to the many figures and different types of data and comparisons it was difficult to find a color scheme for these strains that would be consistent across the manuscript. These colors also reflect the fluorescence markers. We note that not only do we use color to indicate the strain but also text labels.

(4) Panels 3B-F would be best moved to a supplementary figure as this figure is currently very busy. Similarly, I would potentially consider presenting only the bottom row of panels in Panels G-I in the main figure (which would then be consistent with analogous data presented elsewhere).

We have opted to keep these panels in the main text (now Fig. 4) as they are relevant to understanding (1) our justification for why to pursue certain chemoeffector-chemoreceptor interactions and not others, and (2) how the chemoattraction response can be understood both in terms of bacterial population distribution and relevant cells over time.

(5) Fig. 4 and possibly elsewhere - perhaps best not to use Ser as an abbreviation for Serine here because it could potentially be confused with an abbreviation for serum.

It is unfortunate that these two words are so similar. However, Ser is the canonical abbreviation for the amino acid serine. Serum does not have a canonical abbreviation.

(6) Fig. 4 - I would move panels H - K to a separate supplementary figure - currently, they are too squished together and it is hard to make out the x-axis labels. I would also consider moving panels E-G to supplementary as well so that the microscopy images presented elsewhere in the figure can be presented at an appropriate size.

Since we are allowed more figures, we could also break some of these figures up into multiple ones.

(7) Similarly, I would move some panels from Fig. 5 to supplementary as the figure is currently quite busy.

We have rearranged the figure (now Fig. 7) to move the bioinformatics data to Fig. 8 to allow more space for the panels.

Other suggestions(8) Line 179 - how do the concentrations quote for serine and glucose compare to aspartate? This would be helpful to justify the authors' decision not to investigate Tar as a potential chemoreceptor.

This is addressed in our comments above and in Fig. 4A and Fig. 4B-F. Human serum L-Asp is much lower concentration (about 20-fold).

(9) Line 282 - Serine levels in serum are quantified at 241 uM, but this is only discussed in the context of serum growth effects. Could this information be better used to design/ inform the serine gradients that were tested in chemotaxis assays?

We tested a wide range of serine concentrations and show even much lower sources of serine than is present in serum is sufficient for chemoattraction. Also, the K1/2 for serine is 105 uM (Fig. S4), which is surpassed by the concentration in serum (Fig. S5).

(10) The word 'potent' in the title might be too vague, especially as the strength of the response varies between strains/species. It may perhaps be more useful to focus on the rapidity/sensitivity of the response. However, presumably the sensitivity of the response will be driven by the sensitivity of the response to serine (which is already known for *E. coli* at least). Also, as noted in the public review, human serum itself is not a chemoattractant so I would consider re-phasing this in the title and elsewhere.

As suggested, and discussed above, we have implemented this change.

(11) Typo line 59 'context of colonizing of a healthy gut'.

Addressed.

(12) Typo line 538 - there is an extra full stop here.

Addressed.

**Reviewer #2:**
(1) This study is well executed and the experiments are clearly presented. These novel chemotaxis assays provide advantages in terms of temporal resolution and the ability to detect responses from small concentrations. That said, it is perhaps not surprising these bacteria respond to serum as it is known to contain high levels of known chemoattractants, serine certainly, but also aspartate. In fact, the bacteria are shown to respond to aspartate and the tsr mutant is still chemotactic. The authors do not adequately support their decision to focus exclusively on the Tsr receptor. Tsr is one of the chemoreceptors responsible for observed attraction to serum, but perhaps, not the receptor. Furthermore, the verification of chemotaxis to serum is a useful finding, but the work does not establish the physiological relevance of the behavior or associate it with any type of disease progression. I would expect that a majority of chemotactic bacteria would be attracted to it under some conditions. Hence the impact of this finding on the chemotaxis or medical fields is uncertain.

We agree that the data we show are mostly mechanistic and further work is required to learn whether this bacterial behavior is relevant in vivo and during infections. We present new data using an ex vivo intestinal model which supports the feasibility of serum taxis mediating invasion of enterohemorrhagic lesions (Fig. 8).

(2) The authors also state that "Our inability to substantiate a structure-function relationship for NE/DHMA signaling indicates these neurotransmitters are not ligands of Tsr." Both norepinephrine (NE) and DHMA have been shown previously by other groups to be strong chemoattractants for *E. coli* (Ec), and this behavior was mediated by Tsr (e.g. single residue changes in the Tsr binding pocket block the response). Given the 82% sequence identity between the Se and Ec Tsr, this finding is unexpected (and potentially quite interesting). To validate this contradictory result the authors should test *E. coli* chemotaxis to DHMA in their assay. It may be possible that Ec responds to NE and DHMA and Se doesn't. However, currently, the data is not strong enough to rule out Tsr as a receptor to these ligands in all cases. At the very least the supporting data for Tsr being a receptor for NE/DHMA needs to be discussed.

Addressed above. The focus of this study is serum attraction and the mechanisms thereof. We never saw any evidence to support the idea that NE/DHMA drives attraction to serum, nor are chemoeffectors for *Salmonella*, and provide these null-results in Data S2.

(3) The authors also determine a crystal structure of the Se Tsr periplasmic ligand binding domain bound to L-Ser and note that the orientation of the ligand is different than that modeled in a previously determined structure of lower resolution. I agree that the SeTsr ligand binding mode in the new structure is well-defined and unambiguous, but I think it is too strong to imply that the pose of the ligand in the previous structure is wrong. The two conformations are in fact quite similar to one another and the resolution of the older structure, is, in my view, insufficient to distinguish them. It is possible that there are real differences between the two structures. The domains do have different sequences and, moreover, the crystal forms and cryo-cooling conditions are different in each case. It's become increasingly apparent that temperature, as manifested in differential cooling conditions here, can affect ligand binding modes. It's also notable that full-length MCPs show negative cooperativity in binding ligands, which is typically lost in the isolated periplasmic domains. Hence ligand binding is sensitive to the environment of a given domain. In short, the current data is not convincing enough to say that a previous "misconception" is being corrected.

Thank you for this comment, which spurred us to investigate this idea more rigorously. As described above we performed new refinements of the *E. coli* structure edited to have the positions of the ligand and ligand-binding site as modeled in our new Tsr structure from *Salmonella* (Fig. 7J). The best model is obtained with these poses. Along with the poor fit of the *E. coli* model to the density, the best interpretations for these positions, for both structures, are as we have modeled them in the *Salmonella* Tsr structures.

Figure suggestions(1) Figure 2 looks busy and unorganized. Fig 2C could be condensed into one image where there are different colored rings coming from the source point that represent different time points.

Addressed above. Fig. 2 has been broken apart to help improve clarity.

(2) What is the second (bottom) graph of 2D? I think only the top graph is necessary.

We have added an explanation to the figure legend that the top graph shows the means and the bottom shows SEM. The plots cannot easily be overlaid.

(3) Similarly, Fig 2E doesn't need to have so many time points. Perhaps 4 at maximum.

As the development of the response over time is a key take-home of the study, we do not wish to reduce the timepoints shown.

(4) The legend for Figure 2F uses the unit 'µM' to mean micrometers but should use 'µm'.

Corrected.

(5) In Figures 2H-J, the lime green text is difficult to read. The word "serum" does not need to be at the top of each panel. I recommend shortening the y-axis titles on the graphs so you can make the graphs themselves larger.

Addressed above.

(6) In Figures 2H-J, I am confused about what is being shown in the inset graph. The legend says it's the AUC for the data shown. However, in the third panel (S. Typhimurium vs. S. Enteriditus) the data appears to be much more disparate than the inset indicates. I don't think that this inset is necessary either.

The point of this inset graph is to quantify the response through integration of the curve, i.e., area under the curve, which is a common way to quantify complex curves and compare responses as single values. We are using this method to calculate statistical significant of the response compared to a null response. We have added further clarification to the figure legend regarding these plots: Inset plots show foldchange AUC of strains in the same experiment relative to an expected baseline of 1 (no change). p-values shown are calculated with an unpaired two-sided t-test comparing the means of the two strains, or one-sided t-test to assess statistical significance in terms of change from 1-fold (stars).

(7) Line 154, change "relevant for" to "observed in".

Changed.

(8) Line 171, according to the Mist4 database, *Salmonella enterica* has seven chemoreceptors. Why are only Tar, Tsr, and Trg mentioned? Why were only Tsr and Trg tested?

Addressed above.

(9) Line 192, be clear that you are referring to genes and not proteins, as italics are used.

Revised to make this distinction clear.

(10) Line 193, have other studies found a Trg deletion strain to be non-chemotactic? If so, cite this source here.

We state that the Trg deletion strain had deficiencies in motility, and also have revised the text to include the clarification that this was not noted in earlier work with this strain: [line 173]: We were surprised to find that the trg strain had deficiencies in swimming motility (data not shown). This was not noted in earlier work but could explain the severe infection disadvantage of this mutant 34. Because motility is a prerequisite for chemotaxis, we chose not to study the trg mutant further, and instead focused our investigations on Tsr.

(11) Why wasn't a Tar deletion mutant also analyzed? The authors say that based on the known composition of serum, serine and glucose are the most abundant. However, the serum does have aspartate at 10s of micromolar concentrations.

Addressed above.

(12) “The Tsr deletion strain still exhibits an obvious chemoattraction to serum. There are other protein(s) involved in chemoattraction to serum but the text does not discuss this.”

Addressed above.

(13) “In Figure 3B-F, the text is very difficult to read even when zoomed in on.”

We have increased the font size of these panels.

(14) “All of the text in Figure 5 is extremely small and difficult to read.”

Addressed above. We split this figure in two to help improve clarity.

(15) “I wonder about the accuracy of the concentration modeling. It seems like there are a lot of variables that could affect the diffusion rates, including the accuracy of the delivery system. Could the concentrations be verified by the dye experiments?”

Addressed above. We provide a new analysis comparing experimental diffusion of A488 dye compared to calculations (Fig. S2).